# Dynein–Dynactin–NuMA clusters generate cortical spindle-pulling forces as a multi-arm ensemble

**Masako Okumura[1], Toyoaki Natsume[2,3], Masato T Kanemaki[2,3], Tomomi Kiyomitsu[1,4]***

[1]Division of Biological Science, Graduate School of Science, Nagoya University, Nagoya, Japan; [2]Division of Molecular Cell Engineering, National Institute of Genetics, Research Organization of Information and Systems, Shizuoka, Japan; [3]Department of Genetics, SOKENDAI, Shizuoka, Japan; [4]Precursory Research for Embryonic Science and Technology (PRESTO) Program, Japan Science and Technology Agency, Saitama, Japan

**Abstract** To position the mitotic spindle within the cell, dynamic plus ends of astral microtubules are pulled by membrane-associated cortical force-generating machinery. However, in contrast to the chromosome-bound kinetochore structure, how the diffusion-prone cortical machinery is organized to generate large spindle-pulling forces remains poorly understood. Here, we develop a light-induced reconstitution system in human cells. We find that induced cortical targeting of NuMA, but not dynein, is sufficient for spindle pulling. This spindle-pulling activity requires dynein-dynactin recruitment by NuMA's N-terminal long arm, dynein-based astral microtubule gliding, and NuMA's direct microtubule-binding activities. Importantly, we demonstrate that cortical NuMA assembles specialized focal structures that cluster multiple force-generating modules to generate cooperative spindle-pulling forces. This clustering activity of NuMA is required for spindle positioning, but not for spindle-pole focusing. We propose that cortical Dynein-Dynactin-NuMA (DDN) clusters act as the core force-generating machinery that organizes a multi-arm ensemble reminiscent of the kinetochore.

DOI: https://doi.org/10.7554/eLife.36559.001

*For correspondence:
kiyomitsu@bio.nagoya-u.ac.jp

**Competing interests:** The authors declare that no competing interests exist.

## Introduction

Forces generated at dynamic plus-ends of microtubules drive directional movement of chromosomes and the mitotic spindle to achieve successful cell division (*Inoué and Salmon, 1995*). During animal mitosis, dynamic plus-ends of microtubules emanating from the spindle interact with two macro-molecular complexes; kinetochores and the cortical force-generating machinery. Kinetochores consist of more than 100 different proteins assembled on centromeric DNA and surround dynamic microtubule plus-ends using multiple fibril-like microtubule-binding proteins and/or ring-like couplers to harness the energy of microtubule depolymerization for chromosome segregation (*Cheeseman, 2014*; *Dimitrova et al., 2016*; *McIntosh et al., 2008*). In contrast, the cortical force-generating machinery assembles on the plasma membrane and pulls on the dynamic plus-ends of astral microtubules to define spindle position and orientation (*Galli and van den Heuvel, 2008*; *Gönczy, 2008*; *Grill and Hyman, 2005*). Spindle positioning determines daughter cell fate by controlling the distribution of polarized cell fate determinants and daughter cell size during both symmetric and asymmetric cell division (*di Pietro et al., 2016*; *Kiyomitsu, 2015*; *Morin and Bellaïche, 2011*; *Williams and Fuchs, 2013*). In metaphase human cells, the cortical machinery consists of evolutionary conserved protein complexes, including the cytoplasmic dynein motor, its binding partner

**eLife digest** Almost every time a cell divides, it must share copies of its genetic material between two new daughter cells. A large molecular machine called the mitotic spindle makes this happen. The spindle is made of protein filaments known as microtubules that radiate out from two points at opposite ends of the cell. Some of these filaments attach to the genetic material in the center of the cell; some extend in the other direction and anchor the spindle to the cell membrane.

The anchoring filaments – also known as astral microtubules – can position the mitotic spindle, which controls whether the cell splits straight down the middle (to give two identically sized cells) or off-center (which gives cells of different sizes). The force required to move the spindle comes from complexes of proteins under the cell membrane that contain a molecular motor called dynein, its partner dynactin, and three other proteins – including one called NuMA. The astral microtubules interact with this force-generating machinery, but it was unclear how these proteins are arranged at the membrane.

One way to explore interactions in a protein complex is to use a light-induced reconstitution system. This technique involves molecules that will bind together whenever a light shines on them. Fusing these molecules with different proteins means that experimenters can control exactly where, and when, those proteins interact.

Okumura et al. have now used a light-induced reconstitution system to understand how the force-generating machinery positions the spindle in human cells. One of the system's molecules was fused to a protein located at the cell membrane; the other was fused to either the dynein motor or NuMA protein. Using light to move dynein around on the membrane did not move the spindle. Yet, changing the position of NuMA, by moving the light, was enough to rotate the spindle inside the cell.

Understanding how these complexes of proteins work increases our understanding of how cells divide. Using the light-induced system to move the spindle could also reveal more about the role of symmetric and asymmetric cell division in organizing tissues. In particular, being able to manipulate the position and size of daughter cells will provide insight into how cell division shapes and maintains tissues during animal development.

DOI: https://doi.org/10.7554/eLife.36559.002

dynactin, and the cortically-anchored NuMA-LGN-Gαi complex (*Figure 1A*) (*Kiyomitsu and Cheeseman, 2012*). Prior work has conceptualized that the cortical complex is distributed along the cell cortex and individually pulls on astral microtubules using dynein-based motility and/or by controlling microtubule dynamics (*Kiyomitsu and Cheeseman, 2012*; *Kotak and Gönczy, 2013*; *Laan et al., 2012*). However, compared to the focal kinetochore structure, how this diffusion-prone membrane-associated complex efficiently captures and pulls on dynamic plus-ends of astral microtubules remains poorly understood. Here, we sought to understand the mechanisms of cortical pulling-force generation by reconstituting a minimal functional unit of the cortical force-generating complex in human cells using a light-induced membrane tethering. We found that cortical targeting of NuMA is sufficient to control spindle position, and that NuMA makes multiple, distinct contributions for spindle pulling through its N-terminal dynein recruitment domain, central long coiled-coil, and C-terminal microtubule-binding domains. In addition, we demonstrate that NuMA assembles focal clusters at the mitotic cell cortex that coordinate multiple dynein-based forces with NuMA's microtubule binding activities. We propose that the cortical Dynein-Dynactin-NuMA clusters (hereafter referred to as the cortical DDN clusters) act as the core spindle-pulling machinery that efficiently captures astral microtubules and generates cooperative pulling forces to position the mitotic spindle.

## Results

### Optogenetic targeting of NuMA to the mitotic cell cortex is sufficient for dynein-dynactin recruitment and spindle pulling

To understand the molecular mechanisms that underlie cortical force generation, we sought to reconstitute a minimal functional unit of the cortical force-generating machinery in human cells using

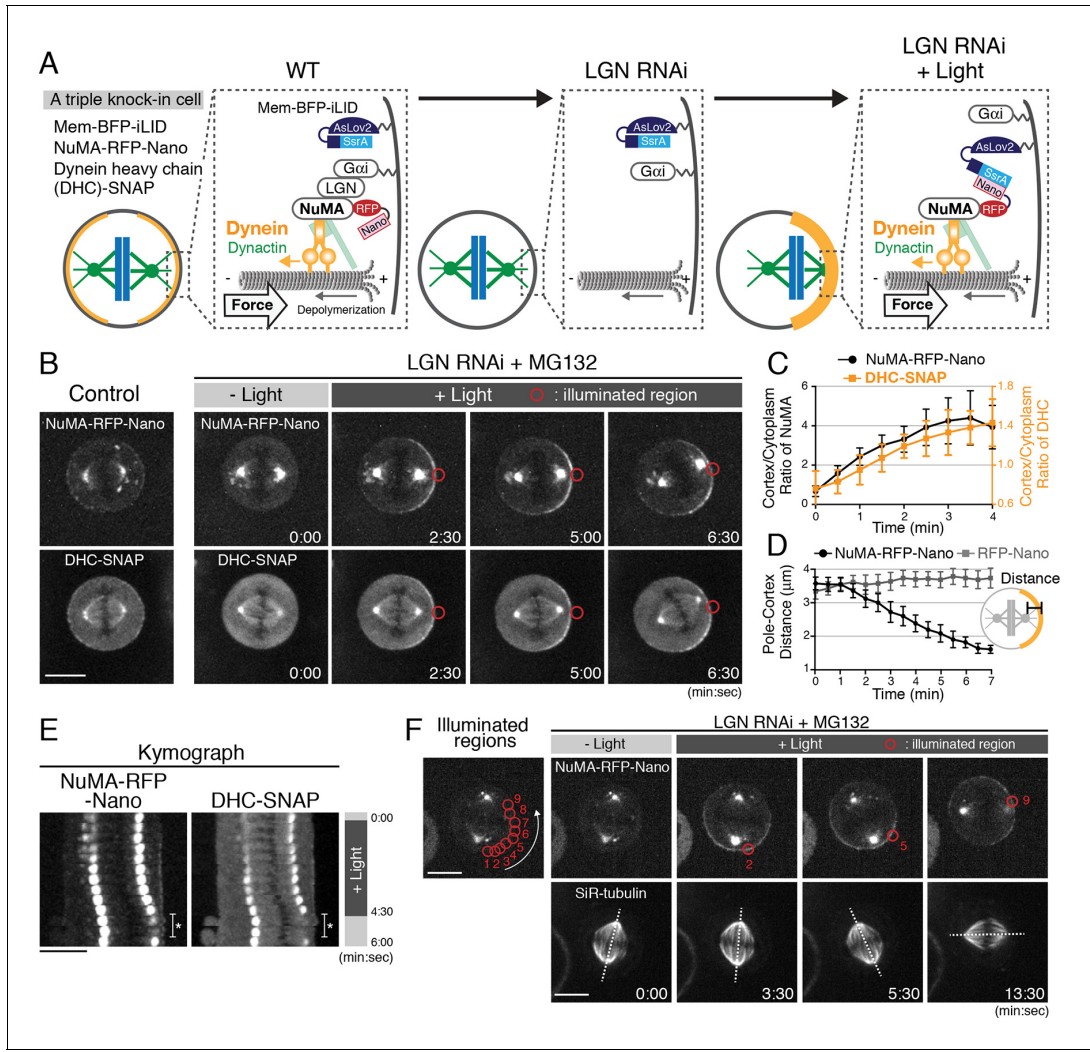

**Figure 1.** Optogenetic targeting of NuMA to the mitotic cell cortex is sufficient for dynein-dynactin recruitment and spindle pulling. (**A**) Diagram summarizing cortical complexes in the indicated conditions. (**B**) Live fluorescent images of NuMA-RFP-Nano (upper) and DHC-SNAP (lower) in control metaphase cells (left), and LGN-depleted cells arrested with MG132. (**C**) Quantification of cortical NuMA-RFP-Nano and DHC-SNAP signals around the light illuminated region (n = 5). Error bars indicate SEM. (**D**) Quantification of the pole-to-cortex distance (NuMA-RFP-Nano, n = 10; RFP-Nano, n = 6). Error bars indicate SEM. (**E**) Kymographs obtained from image sequences in *Figure 1—figure supplement 2A*. Asterisk indicates the duration in which one of the spindle poles moves away from the focal plane. (**F**) When NuMA-RFP-Nano (upper) was optogenetically repositioned at multiple adjacent cortical regions around the cell membrane by sequential illumination (from 1 to 9), the spindle (lower) rotated about 90° in a directed manner coupled with the changes in cortical NuMA enrichment in 55% (n = 11) of cells, but not by repositioning RFP-Nano alone (*Figure 1—figure supplement 2D*, n = 6). Dashed lines indicate the spindle axis. Scale bars = 10 μm.

DOI: https://doi.org/10.7554/eLife.36559.003

The following figure supplements are available for figure 1:

**Figure supplement 1.** Generation of cell lines for light-induced targeting of endogenous NuMA.

DOI: https://doi.org/10.7554/eLife.36559.004

**Figure supplement 2.** Light-induced cortical targeting of NuMA is sufficient for dynein-dynactin recruitment and spindle pulling.

DOI: https://doi.org/10.7554/eLife.36559.005

a light-induced hetero-dimerization system (iLID) (*Guntas et al., 2015*). In this system, cytoplasmic RFP-Nano fusion proteins can be targeted to a locally illuminated region of the mitotic cell cortex by interacting with membrane-bound iLID (*Figure 1A*; *Figure 1—figure supplement 1A–B*; and *Video 1*). Because the N-terminal fragment of NuMA is sufficient to recruit dynein-dynactin to the cell cortex (*Kotak et al., 2012*), we first sought to manipulate endogenous NuMA. We established

triple knock-in cell lines that stably express membrane-targeted BFP-iLID (Mem-BFP-iLID), a NuMA-RFP-Nano fusion (*Figure 1A*; *Figure 1—figure supplement 1C–E*), and SNAP-tagged dynein heavy chain (DHC) or the dynactin subunit p150 (*Figure 1—figure supplement 1F–G*). To prevent cortical recruitment of NuMA by the endogenous LGN-Gαi complex, we depleted LGN by RNAi (*Figure 1A* middle, 1B t = 0:00; *Figure 1—figure supplement 1H*). We then continuously illuminated the cortical region next to one of spindle poles (indicated by red circles in Figures) with a 488 nm laser to induce NuMA-RFP-Nano targeting. Light illumination induced the asymmetric cortical accumulation of NuMA-RFP-Nano within a few minutes (*Figure 1B–C*), which subsequently recruited DHC-SNAP and p150-SNAP (*Figure 1B–C*; *Figure 1—figure supplement 2A–C*). The level of light-induced cortical NuMA is about three times higher than that of endogenous NuMA in metaphase, but similar to that in anaphase (*Figure 1—figure supplement 1I–J*).

Importantly, following asymmetric NuMA-RFP-Nano targeting, the mitotic spindle was gradually displaced toward the light-illuminated region in 82.4% of cells (n = 17, *Figure 1B,D–E*, and *Video 2*), whereas spindle displacement and cortical dynein recruitment was never observed by targeting RFP-Nano alone (n = 6, *Figure 1D* and *Figure 1—figure supplement 2D*). Additionally, we found that light-induced repositioning of cortical NuMA is sufficient to drive spindle rotational re-orientation (*Figure 1F* and *Video 3*), and that light-induced NuMA targeting also causes spindle displacement in 71.4% of Gαi (1 + 2 + 3) depleted cells (n = 7, *Figure 1—figure supplement 2E–F*). These results indicate that light-induced cortical recruitment of the Dynein-Dynactin-NuMA (DDN) complex is sufficient, and that LGN/Gαi are dispensable for generating cortical spindle-pulling forces in human cells.

## Light-induced cortical DDN complex can pull on taxol-stabilized astral microtubules

Cortical pulling forces are supposed to be generated by dynein-based motility on astral microtubules and/or astral microtubule depolymerization coupled with cortical anchorage (*Grill and Hyman, 2005*). To understand the contributions of astral microtubules to the spindle movement caused by light-induced cortical NuMA, we disrupted or stabilized astral microtubules using the microtubule-targeting drugs, nocodazole or taxol, respectively. In control cells, the metaphase spindle contains visible astral microtubules (*Figure 2A*, left) and is displaced following light-induced NuMA-RFP-Nano targeting (*Figure 2B,D–E*). In contrast, when astral microtubules were selectively disrupted by treatment with 30 nM nocodazole (*Figure 2A*, middle), the spindle was no longer displaced in 56% of cells (n = 5/9 cells), and only partially displaced in the remaining 44% of cells (n = 4/9) (*Figure 2C–E*), despite presence of cortical dynein (*Figure 2C* t = 5:30). This suggests that astral microtubules are required for spindle pulling by the light-induced cortical DDN complex.

Treatment with 10 µM taxol stabilized astral microtubules based on increases in both the length and number of astral microtubules 1 min after addition of taxol (*Figure 2A*, right) (*Rankin and Wordeman, 2010*). Importantly, even in the presence of 10 µM taxol, the spindle was gradually displaced toward the light-illuminated region (*Figure 2H-G*, t = 5:00). In these taxol-treated cells, the velocity of spindle movement was slower than that observed in control cells (*Figure 2F–I*), suggesting that depolymerization of astral microtubules may also contribute to force generation, although this reduced velocity might be caused alternatively by cortical pushing by stabilized astral microtubules. In these experiments, we visualized spindle microtubules with 50 nM SiR-tubulin (*Lukinavičius et al., 2014*), a fluorescent docetaxel derivative, and confirmed the presence of 10 µM taxol by the decrease of SiR-tubulin intensity (*Figure 2G* t = 0:00), likely due to competition for the same microtubule-binding site. These results suggest that the light-induced cortical DDN complex generates

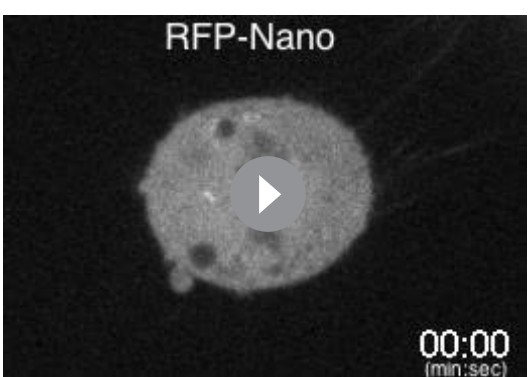

**Video 1.** Light-induced cortical targeting of RFP-Nano. The dynamic cortical targeting and repositioning of RFP-Nano, in response to illuminations, are shown in this movie; it is played at five fps.
DOI: https://doi.org/10.7554/eLife.36559.006

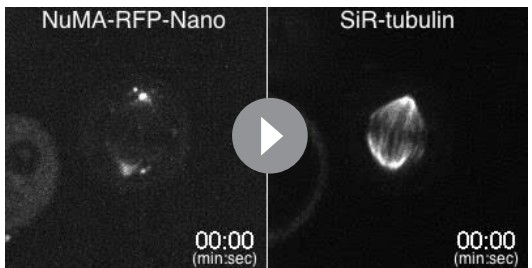

**Video 2.** Light-induced cortical targeting of NuMA-RFP-Nano and spindle pulling. Light-induced cortical recruitment of NuMA-RFP-Nano (left), and DHC-SNAP (right), and spindle displacement toward NuMA/DHC-enriched cell cortex have been shown in this movie; it is played at five fps.
DOI: https://doi.org/10.7554/eLife.36559.007

**Video 3.** Light-induced cortical repositioning of NuMA-RFP-Nano and spindle rotation. Light-induced cortical repositioning of NuMA-RFP-Nano (left), and dynamics of SiR-tubulin (right) have been shown in this movie. The spindle rotation was coupled with cortical repositioning of NuMA. This movie is played at five fps.
DOI: https://doi.org/10.7554/eLife.36559.008

cortical pulling forces by using dynein-based motility on astral microtubules even if microtubule depolymerization is inhibited.

## Dynein activity is required for spindle displacement by the cortical DDN complex

Recently, ciliobrevin D was developed as a specific dynein inhibitor (*Firestone et al., 2012*). This compound inhibits dynein's microtubule gliding and ATPase activity, but not the association between ADP-bound dynein and microtubules in vitro. To understand the requirement of these dynein activities for force generation, we next sought to analyze spindle displacement following ciliobrevin D treatment. In HCT116 cells, we found that ciliobrevin D treatment in interphase caused mitotic phenotypes including chromosome misalignment similar to dynein degradation (*Natsume et al., 2016*) under 0.5%, but not 10%, FBS culture conditions (*Figure 3—figure supplement 1A–D*), consistent with a previous report (*Firestone et al., 2012*). We next added ciliobrevin D in metaphase-arrested cells. Although dynein activity is required to maintain spindle bipolarity, we found that spindle bipolarity was maintained for ~30 min following the treatment of ciliobrevin D, and was gradually disrupted during the subsequent 30–60 min (*Figure 3—figure supplement 1E–G*). We next performed the optogenetic spindle-pulling assay during the initial 60 min of ciliobrevin treatment according to the Procedure depicted in *Figure 3A*. In control cells, light-induced targeting of NuMA displaced the spindle in 80% of cell (n = 10, *Figure 3B and D*). In contrast, the spindle was not displaced in 75% of ciliobrevin D-treated cells (n = 12, *Figure 3C–D*), whereas dynein was normally recruited to the cell cortex and the bipolar spindle structure was maintained during the assay. These results suggest that light-induced NuMA not only recruits, but also activates dynein at the cell cortex to generate cortical pulling forces.

## Light-induced cortical targeting of dynein is not sufficient to pull on the spindle in human cells

A dimerized version of the yeast dynein motor domain is sufficient to position microtubule asters in microfabricated chambers (*Laan et al., 2012*). To understand the sufficiency of cortical dynein for generating spindle-pulling forces within a human cell, we next directly targeted dynein to the cell cortex (*Figure 3E*). Similar to the NuMA-RFP-Nano fusion, endogenously tagged Nano-mCherry-DHC asymmetrically accumulated at the light-illuminated region within several minutes (*Figure 3F*; *Figure 3—figure supplement 1H*), and subsequently recruited SNAP-tagged endogenous p150/dynactin to this cortical region (*Figure 3F–G*; *Figure 3—figure supplement 1I*). However, endogenous NuMA-SNAP was not recruited to the light illuminated region (*Figure 3H*; *Figure 3—figure supplement 1I*), and the spindle was not displaced toward dynein/dynactin-enriched cortex (*Figure 3F* right, and *Figure 3I*) despite the fact that substantial levels of dynein were recruited to the cell cortex (compare *Figure 3G* to *Figure 1C*). These results suggest that cortical dynein targeting is not sufficient for generating cortical pulling forces in human cells, consistent with recent

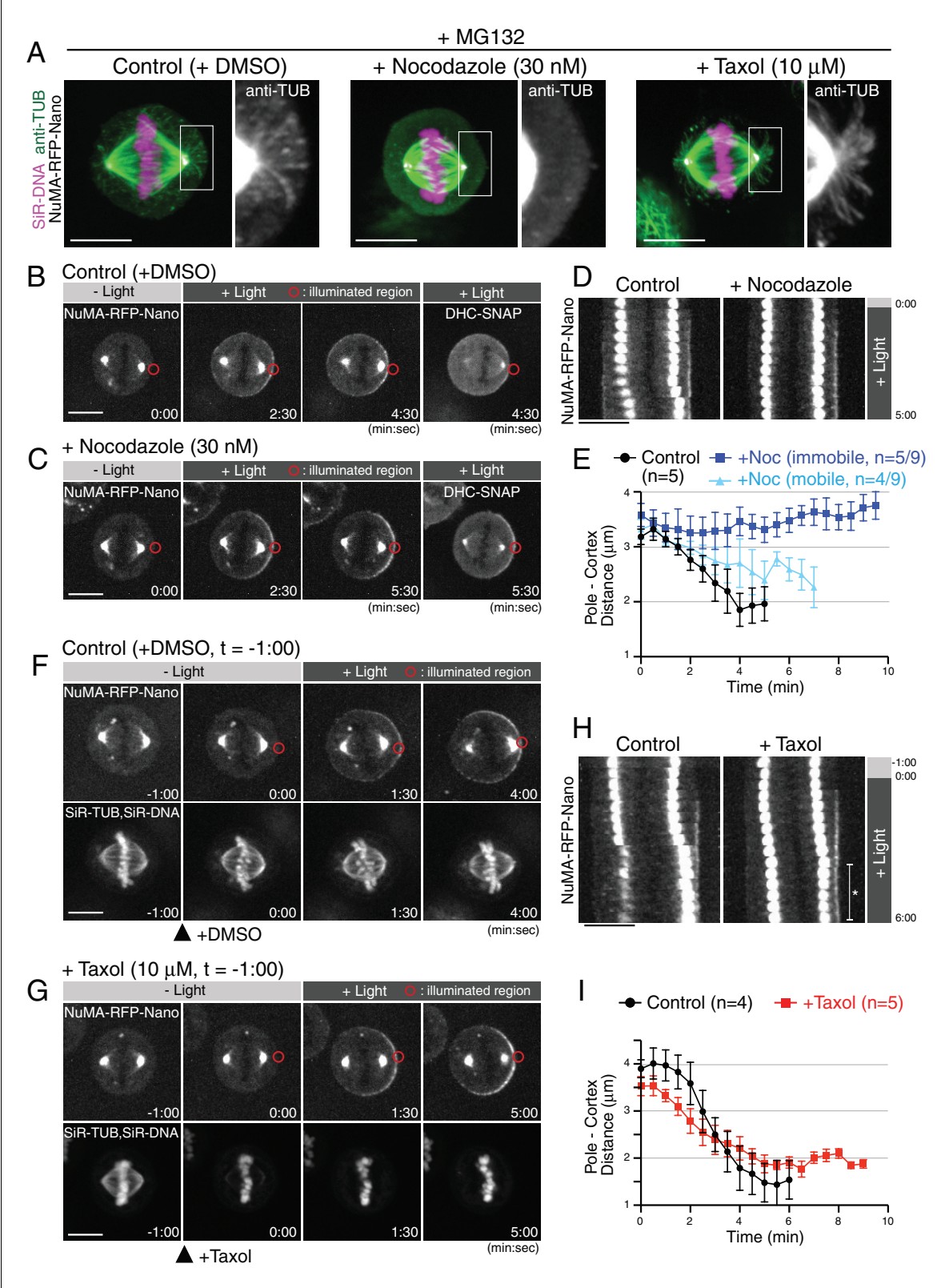

**Figure 2.** Light-induced cortical NuMA-dynein complex pulls on taxol-stabilized astral microtubules. (**A**) Fluorescent images of astral microtubules in fixed HCT116 cells treated with drugs as indicated. Cells were arrested at metaphase with MG132 for 1 hr, and DMSO/nocodazole or taxol were then added for 30 or 1 min, respectively. Images are maximally projected from 15 z-sections acquired using 0.2 μm spacing. (**B** and **C**) Live fluorescent images of NuMA-RFP-Nano (upper) and DHC-SNAP (lower) treated with DMSO (**B**) or nocodazole (**C**). (**D**) Kymographs obtained from image sequences

*Figure 2 continued on next page*

*Figure 2 continued*

in (**B**) and (**C**) showing the movement of the spindle at 30 s intervals. (**E**) Pole-to-cortex distance for control (black, n = 5), and nocodazole-treated cells (blue or light-blue). Blue and light-blue graphs indicate immobile (n = 5/9) and partially mobile pools (n = 4/9), respectively. Error bars indicate SEM. (**F** and **G**) Live fluorescent images of NuMA-RFP-Nano (upper), and SiR-tubulin and SiR-DNA (*Lukinavičius et al., 2015*) (lower), treated with DMSO (**F**) or taxol (**G**). DMSO and Taxol were added at −1:00, and light illumination began at 0:00, when SiR-tubulin images were selectively abolished by taxol treatment. (**H**) Kymographs obtained from image sequences in (**F**) and (**G**) at 30 s intervals. In taxol-treated cells, the spindle did not attach to the cell cortex as indicated with an asterisk, likely due to stabilized astral microtubules. (**I**) Pole-to-cortex distance for control (black, n = 4), and taxol-treated cells (red, n = 5). Error bars indicate SEM. Scale bars = 10 μm.

DOI: https://doi.org/10.7554/eLife.36559.009

studies demonstrating that human dynein is auto-inhibited (*Torisawa et al., 2014*; *Zhang et al., 2017*) and dynactin and cargo adaptors are required to activate dynein motility (*McKenney et al., 2014*; *Schlager et al., 2014*; *Zhang et al., 2017*). Although we cannot exclude the possibility that iLID-Nano mediated cortical targeting of DHC may impair some cortical dynein functions or assemblies in human cells, cortical dynein anchoring with ePDZ-LOVp system in *C. elegans* is also insufficient to generate cortical pulling forces (*Fielmich et al., 2018*).

## A Spindly-like motif in NuMA is required for cortical dynein recruitment, but not sufficient for spindle pulling

The above results suggest that NuMA is required to activate dynein at the cell cortex. Thus, we next sought to define the minimal functional region of NuMA as a dynein adaptor (*Figure 4A*). Importantly, our truncation analyses revealed that the NuMA N-terminal region contains a Spindly-like motif sequence (*Figure 4B–E*; *Figure 4—figure supplement 1A–G*) that was recently identified as a conserved binding motif for the pointed-end complex of dynactin in dynein cargo adaptors (*Gama et al., 2017*). We found that NuMA wild type (WT) fragment (1-705), but not a Spindly-motif (SpM) mutant containing alanine mutations in the Spindly-motif (*Figure 4D*), recruited dynein to the light-illuminated cortical region (*Figure 4F* and *Figure 4—figure supplement 1H*). However, the NuMA (1-705) WT and longer NuMA (1–1700) fragments were unable to fully displace the spindle despite the presence of substantial levels of cortical dynein (*Figure 4B–C,H–I*; *Figure 4—figure supplement 1I–L*), whereas ectopically expressed full length NuMA (1–2115 ΔNLS) was able to displace the spindle in ~40% of cells (*Figure 4G*; the NLS was deleted to eliminate dimerization with endogenous NuMA by spatially separating exogenously expressed constructs from the nuclear-localized endogenous NuMA before G2 release. In contrast, exogenously expressed NLS containing NuMA-RFP-Nano (1–2115) accumulated in the nucleus before G2, but was unable to displace the spindle efficiently (11.1%, n = 9), likely due to weak cortical anchorage by hetero-dimerization with endogenous NuMA lacking RFP-Nano). These results suggest that NuMA recruits dynein-dynactin via its N-terminal Spindly motif, likely to activate dynein's motility at the mitotic cell cortex similarly to other dynein cargo adaptors (*Gama et al., 2017*; *McKenney et al., 2014*; *Schlager et al., 2014*). However, despite this activation, additional NuMA domains are required to produce cortical spindle-pulling forces.

## NuMA's C-terminal microtubule-binding domains are required for spindle pulling

At kinetochores, a multiplicity of microtubule-binding activities is required to generate cooperative pulling forces (*Cheeseman et al., 2006*; *Schmidt et al., 2012*). Because NuMA's C-terminal region contains two microtubule-binding domains (MTBD1, and MTBD2) (*Figure 5A* and *Figure 5—figure supplement 1A*) (*Chang et al., 2017*; *Du et al., 2002*; *Gallini et al., 2016*; *Haren and Merdes, 2002*), direct binding of NuMA to astral microtubules may generate cooperative forces in parallel with dynein-dynactin recruitment as recently proposed by Seldin et al (*Seldin et al., 2016*). Consistent with this, we found that a Nano fusion with a NuMA (1–1895) fragment, which lacks both microtubule-binding domains, was unable to fully displace the spindle regardless of cortical dynein recruitment (*Figure 5B–C*; *Figure 5—figure supplement 1B*). Similarly, NuMA (1–1985), which lacks only the C-terminal microtubule-binding domain (MTBD2), was unable to displace the spindle (*Figure 5B,D*; *Figure 5—figure supplement 1C*). In contrast, NuMA Δex24, which lacks exon 24 thus disrupting MTBD1 and an NLS (*Figure 5A*) (*Gallini et al., 2016*; *Seldin et al., 2016*; *Silk et al.,*

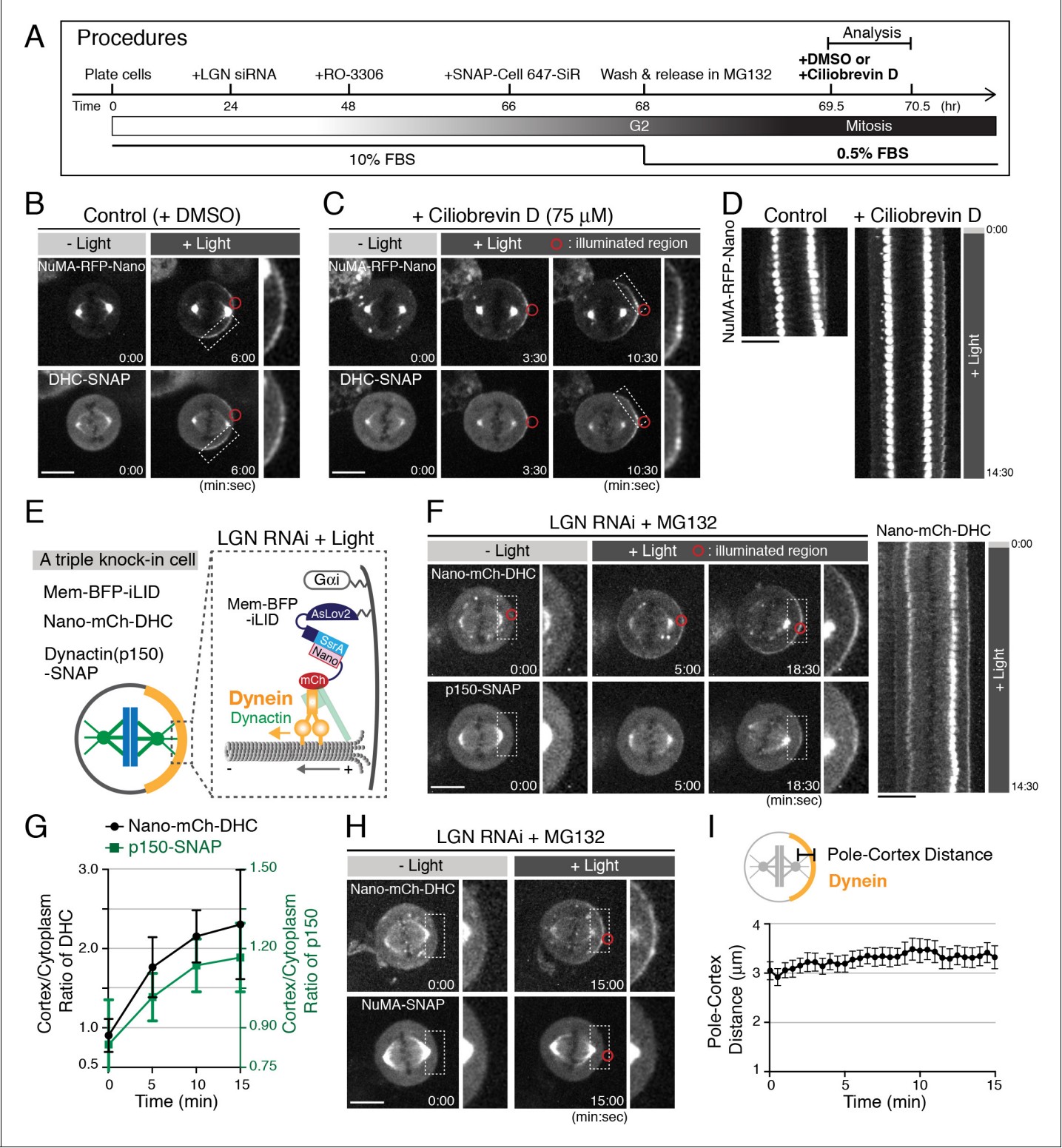

**Figure 3.** Dynein activity is required for spindle pulling, but light-induced cortical dynein targeting is not sufficient to pull on the spindle. (A) Schematic of experimental procedures. The FBS concentration in the culture medium was changed from 10 to 0.5% at the 68 hr mark. DMSO or ciliobrevin D was added at the 69.5 hr mark and the cells were analyzed for 1 hr. (B and C) Live fluorescent images of NuMA-RFP-Nano (upper) and DHC-SNAP (lower) treated with DMSO (B) or ciliobrevin D (C). (D) Kymographs obtained from image sequences in (B) and (C) showing the movement of the spindle at 30 s intervals. (E) Cortical complexes formed by light-induced targeting of Nano-mCherry-DHC. (F) Left: live fluorescent images of Nano-mCherry-DHC (upper) and p150-SNAP (lower). Right: kymograph obtained from image sequences on the left. (G) Quantification of cortical Nano-mCherry-DHC and

*Figure 3 continued on next page*

Figure 3 continued

p150-SNAP signals around the light illuminated region (n = 6). Error bars indicate SEM. (H) Live fluorescent images of Nano-mCherry-DHC (upper) and NuMA-SNAP (lower). (I) Measurement of the pole-to-cortex distance (n = 10). Error bars indicate SEM. Scale bars = 10 μm.

DOI: https://doi.org/10.7554/eLife.36559.010

The following figure supplement is available for figure 3:

**Figure supplement 1.** Generation of knock-in cell lines for the DHC and mitotic phenotypes caused by ciliobrevin D treatment.

DOI: https://doi.org/10.7554/eLife.36559.011

*2009*), was able to recruit dynein and displace the spindle similarly to the NuMA-ΔNLS construct (*Figure 5B,E–F*; *Figure 4—figure supplement 1J*). Because the corresponding mouse NuMA Δex22 mutant shows spindle orientation defects in mouse keratinocytes and the epidermis (*Seldin et al., 2016*), this region may have specific roles in different cell types. Alternatively, weak defects in the NuMA Δex24 mutant may be suppressed by targeting increased levels of cortical NuMA Δex24 in this assay. These results indicate that NuMA's microtubule binding domains, particularly MTBD2, play critical roles for the ability of the DDN complex to generate spindle-pulling forces.

## NuMA's central coiled-coil is required for pulling on the spindle

The work described above defines two important molecular features for cortical force generation: dynein recruitment/activation through the Spindly-like motif and a distinct direct microtubule-binding activity by NuMA. To test whether these features are sufficient to generate cortical pulling forces, we next expressed a fusion construct, NuMA (N + C ΔNLS), that contains both its dynein-recruiting N-terminal and microtubule-binding C-terminal domains, but lacks a ~1000 aa region of its central coiled-coil (*Figure 5A* #12). The NuMA fusion, but not the C-terminal domain (1700–2115) alone (NuMA-C), recruited DHC-SNAP to the light-illuminated region (*Figure 5G–I*; *Figure 5—figure supplement 1D*). However, the NuMA (N + C ΔNLS) fusion was unable to fully displace the spindle (*Figure 5G*; *Figure 4—figure supplement 1J*). These results indicate that NuMA's 200 nm long, central coiled-coil (*Harborth et al., 1999*) also functions with its N-terminal and C-terminal domains to efficiently capture and pull on astral microtubules.

## Identification of a clustering domain on NuMA's C-terminal region

Our results reveal that NuMA has multiple functional modules for force generation. However, considering the sophisticated kinetochore structure that surrounds a plus-end of microtubule with multiple microtubule-binding proteins (*Cheeseman, 2014*; *Dimitrova et al., 2016*), we next sought to define the architecture of the cortical attachment site that is required to efficiently capture and pull on dynamic plus-ends of astral microtubules. Intriguingly, we found that NuMA constructs containing its C-terminal region displayed punctate cortical signals, which tended to be even more evident in smaller constructs (e.g. *Figure 5H–I*). These results suggest that NuMA forms oligomeric structures at the mitotic cell cortex as observed in vitro (*Harborth et al., 1999*). To understand mechanisms of the NuMA's C-terminal oligomerization/clustering at the mitotic cell cortex, we took advantages of a NuMA-C 3A fragment, which eliminates CDK phosphorylation sites (*Compton and Luo, 1995*) allowing NuMA to localize to the metaphase cell cortex independently of LGN (*Kiyomitsu and Cheeseman, 2013*). Similar to the NuMA-C-RFP-Nano (*Figure 5I*), GFP-NuMA-C 3A displayed punctate cortical signals (*Figure 6A–B* #C1), which was distinct from that of its cortical interacting partners - phospholipids and 4.1 proteins (*Kiyomitsu and Cheeseman, 2013*; *Kotak et al., 2014*; *Mattagajasingh et al., 2009*; *Zheng et al., 2014*) – that localize homogenously to the cell cortex (*Figure 6—figure supplement 1A–B*). Interestingly, the punctate NuMA-C 3A patterns intercalated with cortical actin localization, and still localized following the disruption of actin polymerization (*Figure 6—figure supplement 1C*). These results suggest that the NuMA C-terminal fragment self-assembles on the membrane independently of its cortical binding partners and actin cytoskeleton.

Importantly, by analyzing different truncations and mutants, we found that a 100 aa region (aa: 1700–1801) of NuMA adjacent to its 4.1 binding domain is required for the formation of punctate foci (*Figure 6A–B*, compare #C1 to #C2), and further that a highly conserved 10 amino acid region, E1768-P1777 (*Figure 6C*), is necessary for cluster formation (*Figure 6B*, see 5A-2 and 5A-3 alanine mutants; *Figure 6—figure supplement 1D–F*). Consistently, the 1700–1895 region of NuMA is

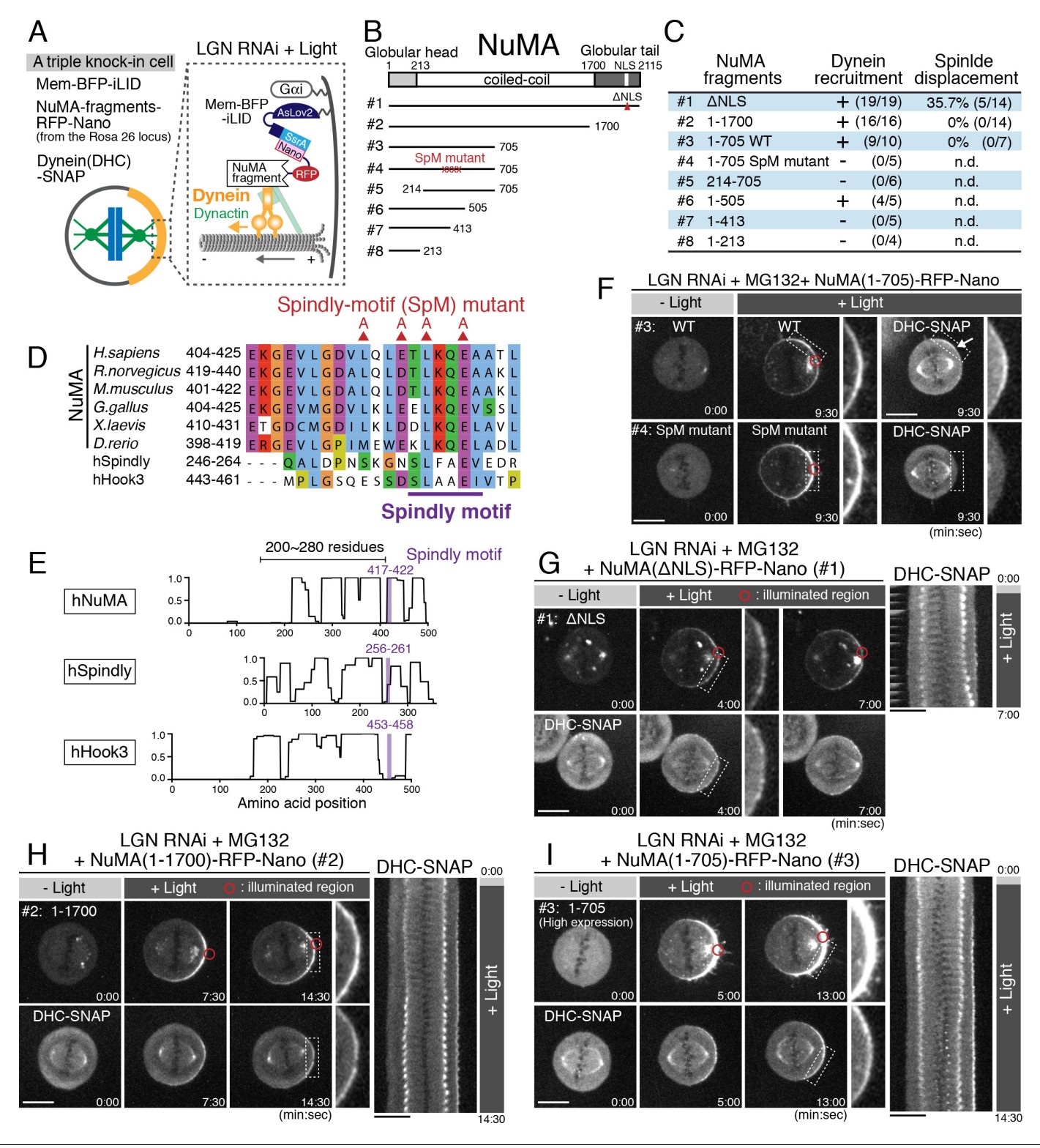

**Figure 4.** A Spindly-like motif in NuMA is required for cortical dynein recruitment, but not sufficient for spindle pulling. (**A**) Cortical complexes formed by light-induced targeting of NuMA fragments fused with RFP-Nano. (**B**) Full-length NuMA and the tested NuMA truncation fragments. Globular domains at N- and C-terminal regions of NuMA are indicated in light-gray and gray, respectively. (**C**) A summary of the frequency of cortical dynein recruitment and spindle displacement by targeted constructs. See *Figure 4—figure supplement 1I–J* for details. (**D**) Amino acid sequence alignment of the Spindly-motif like region of NuMA proteins in *H. Sapiens* (NP_006176), *R. norvegicus* (NP_001094161), *M. musculus* (NP_598708), *G. gallus*

*Figure 4 continued on next page*

*Figure 4 continued*

(NP_001177854), *X. laevis* (NP_001081559), *D. rerio* (NP_001316910), and human Spindly (NP_001316568) and Hook3 (NP_115786) aligned by ClustalWS. The conserved L and E substituted by alanine are indicated with red triangles. (E) Lupas coils prediction (window 21). Spindly motif (purple) is commonly located at the C-terminal region of the coiled-coil, with 200 ~ 280 residues. (F) Live fluorescent images of NuMA (1-705) WT (upper) and SpM mutant (lower). DHC-SNAP images are shown to the right. (G–I) Left: live fluorescent images of NuMA constructs (upper) and DHC-SNAP (lower). Right: kymographs obtained from image sequences of DHC-SNAP on the left at 30 s intervals. Scale bars = 10 μm.
DOI: https://doi.org/10.7554/eLife.36559.012
The following figure supplement is available for figure 4:

**Figure supplement 1.** The N-terminal region of NuMA is required for cortical dynein recruitment.
DOI: https://doi.org/10.7554/eLife.36559.013

required for the NuMA fragments to display punctate cortical signals (compare *Figure 4H* to *Figure 5C*; *Figure 6—figure supplement 1G*). These results suggest an exciting possibility that NuMA assembles a specialized structure to produce large spindle-pulling forces at the cell cortex.

## Clustering by NuMA is required for spindle pulling and positioning, but not for spindle-pole focusing

Above we identified NuMA mutants (5A-2, 5A-3) that are unable to form clusters at the mitotic cell cortex (*Figure 6B–C*). To test the functional importance of the novel clustering behavior of NuMA, we next analyzed cortical force generation by full length NuMA wild-type (WT) compared to the 5A-3 mutant using Nano fusions. In cells expressing NuMA (1–2115 ΔNLS)-RFP-Nano (WT), NuMA and DHC-SNAP became gradually detectable as punctate foci (*Figures 6D*, 4:30 and 11:00), and the spindle was displaced towards the light-illuminated region (*Figures 6D*, 13:00). In contrast, when the NuMA 5A-3 mutant was targeted to the cell cortex, both NuMA 5A-3 mutant and DHC failed to form punctate foci (*Figure 6E*; *Figure 6—figure supplement 1H*), similarly to GFP-NuMA-C 5A-3 (*Figure 6B*), and the spindle was not fully displaced (*Figure 5B* #14, *Figure 6E*; *Figure 6—figure supplement 1H*). These results indicate that NuMA's clustering activity correlates with the generation of cortical pulling forces.

To further probe functional importance of the NuMA's clustering activity, we next replaced endogenous NuMA with either NuMA WT or the 5A-3 mutant using the auxin-induced degron (AID) system (*Figure 7A*) (*Natsume et al., 2016*). Consistent with the above results, endogenous NuMA fused to mAID-mClover-FLAG tag (NuMA-mACF) displayed punctate cortical signals that colocalized with dotted signals of SNAP-tagged dynein and LGN (*Figure 7—figure supplement 1A–C*). When the endogenous NuMA-mACF was degraded, 80% of mitotic cells (n = 63) displayed abnormal spindles with unfocused microtubules (*Figure 7B* #2; *Figure 7—figure supplement 1D–E*), consistent with the NuMA KO phenotypes in human hTERT-RPE1 cells (*Hueschen et al., 2017*). However, both NuMA WT and the 5A-3 mutant were able to rescue these abnormal spindle phenotypes (*Figure 7B* #3 and #4), suggesting that clustering of NuMA is dispensable for microtubule focusing at the spindle poles. In contrast, when endogenous NuMA was replaced with NuMA 5A-3 mutant, the metaphase spindle was tilted and randomly oriented on the x-z plane (26.8 ± 20.7°, n = 37, *Figure 7C*; *Figure 7—figure supplement 1F*) whereas the spindle in NuMA WT cells was oriented parallel to the substrate (10.7 ± 9.6°, n = 34, *Figure 7C*) as observed in control metaphase cells (11.5 ± 11.8°, n = 41, *Figure 7C*). These results suggest that NuMA's C-terminal clustering is required for proper spindle orientation. We note that the 5A-3 mutation site contains Y1774 (*Figure 6C*), which is phosphorylated by ABL1 kinase and contributes to proper spindle orientation (*Matsumura et al., 2012*). However, treatment with the ABL1 kinase inhibitor Imatinib caused only a mild spindle orientation phenotype (12.3 ± 14.7°, n = 27, *Figure 7C*), suggesting that the spindle mis-orientation phenotype observed in the 5A-3 mutant is largely attributable to defects in NuMA clustering. Taken together, these results indicate that clustering activity of NuMA is required at the mitotic cell cortex, but not at the spindle poles, for generating cortical pulling forces. Thus, NuMA has a location-dependent structural function that clusters multiple DDN complexes to efficiently capture and pull on dynamic plus ends of astral microtubules.

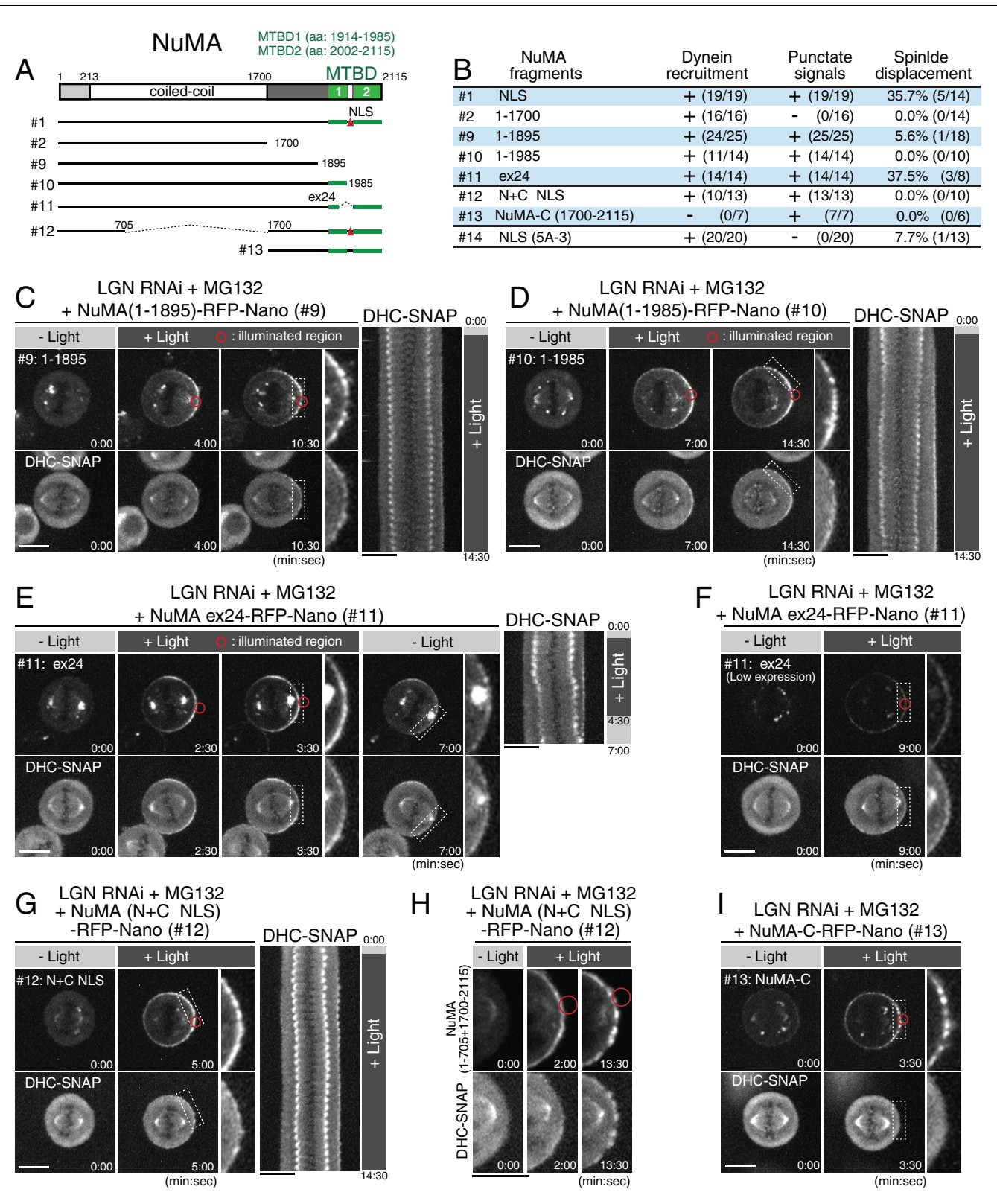

**Figure 5.** NuMA's C-terminal microtubule-binding domains and central coiled-coil are required for spindle pulling. (**A**) Full-length NuMA and the tested NuMA truncation fragments. Microtubule binding domains (MTBDs) are indicated in green. (**B**) Summary of the frequency of cortical dynein recruitment, dot signal formation and spindle displacement by targeted constructs. *Figure 4—figure supplement 1I–J* for details. (**C–E**) Left: live fluorescent images of indicated NuMA constructs (upper) and DHC-SNAP (lower). Right: kymographs obtained from image sequences of DHC-SNAP

*Figure 5 continued on next page*

*Figure 5 continued*
on the left at 30 s intervals. (F) Live fluorescent images of NuMA Δex24-RFP-Nano (upper) and DHC-SNAP (lower). Expression level of NuMA Δex24-RFP-Nano was lower than that in (E), but the spindle was still displaced. (G) Left: live fluorescent images of NuMA (N + C ΔNLS)-RFP-Nano (upper) and DHC-SNAP (lower). Right: kymographs obtained from image sequences of DHC-SNAP on the left at 30 s intervals. (H) Enlarged images of NuMA (N + C ΔNLS)-RFP-Nano (upper) and DHC-SNAP (lower) at indicated times. (I) Live fluorescent images of NuMA-C-RFP-Nano (upper) and DHC-SNAP (lower). Scale bars = 10 μm.

DOI: https://doi.org/10.7554/eLife.36559.014
The following figure supplement is available for figure 5:

**Figure supplement 1.** Light-induced targeting of exogenously expressed NuMA fragments lacking C-terminal MTBDs and central coiled-coil.
DOI: https://doi.org/10.7554/eLife.36559.015

## Discussion

### The cortical DDN complex acts as a core functional unit of the cortical force-generating machinery

Here, we applied a light-induced targeting system, iLID (*Guntas et al., 2015*), for *in cell* reconstitution of the cortical force-generating machinery (e.g. *Figure 1A–B*). Our work demonstrates that light-induced targeting of NuMA, but not dynein, is sufficient to control spindle position and orientation in human cells. This is consistent with recent findings that mammalian dynein requires cargo adaptors to activate its motility in vitro (*McKenney et al., 2014*; *Schlager et al., 2014*; *Zhang et al., 2017*). In addition, our findings suggest that LGN/Gαi are dispensable for force generation, and instead act as receptors that specify the position of NuMA at the cell membrane. Consistent with this model, LGN-independent pathways that target NuMA to the cell cortex have been reported, such as Dishevelled (*Ségalen et al., 2010*) and phospho-lipids (*Zheng et al., 2014*). Thus, we propose that the Dynein-Dynactin-NuMA (DDN) complex is a universal core unit that constitutes the cortical force-generating machinery, whereas LGN and other receptors specify the targeting of the DDN complex to the membrane.

### NuMA acts as a force-amplifying platform at the mitotic cell cortex

Our work demonstrates four distinct functions for NuMA at the mitotic cell cortex. First, NuMA recruits dynein-dynactin through its N-terminal region. We found that the conserved Spindly-like motif in NuMA is required for dynein recruitment (*Figure 4D–F*). NuMA may directly interact with the dynactin pointed-end complex through this Spindly-like motif similarly to other dynein cargo adaptors (*Gama et al., 2017*), and activate dynein motility at the mitotic cell cortex. Second, the central long coiled-coil of NuMA is required for spindle pulling (*Figure 5G*). Purified NuMA displays a long (~200 nm) rod-shaped structure that shows flexibility with a main flexible-linker region near the middle of central coiled coil (*Harborth et al., 1999*). Longer flexible arms of NuMA may increase the efficiency of astral microtubule capture by the dynein-dynactin complex, similarly to fibril-like Ndc80 complexes and CENP-E motors at kinetochores (*Kim et al., 2008*; *McIntosh et al., 2008*). Third, NuMA contributes to cortical force generation with its own C-terminal microtubule-binding domains (MTBDs) (*Figure 5C*), particularly MTBD2 (*Figure 5D*). Because this region is also required to prevent hyper-clustering (*Figure 5—figure supplement 1C* right), and is sufficient for cortical localization in anaphase (*Figure 6—figure supplement 1F* C#3, T.K. unpublished observation), this region may play multiple roles for cortical pulling-force generation. Interestingly, a NuMA C-terminal fragment containing MTBD1 (aa: 1811–1985, called NuMA-TIP, *Figure 5—figure supplement 1A*) accumulates at microtubule tips, and remains associated with stalled and/or deploymerizing microtubules (*Seldin et al., 2016*). By using its two microtubule-binding domains, NuMA may harness the energy of microtubule depolymerization for pulling on astral microtubules similar to the human Ska1 complex or yeast Dam1 ring complex at kinetochores, both of which track with depolymerizing microtubules (*Schmidt et al., 2012*; *Westermann et al., 2006*).

Finally, we demonstrate that NuMA generates large pulling forces by clustering the DDN complexes through its C-terminal clustering domain (*Figure 6C–E*), similar to lipid microdomains on phagosomes that achieve cooperative force generation of dynein (*Rai et al., 2016*). Previous studies demonstrated that the 1700–2003 region of NuMA is required for oligomerization in vitro (*Harborth et al., 1999*). We defined the 1700–1801 region of NuMA as a clustering domain (CD)

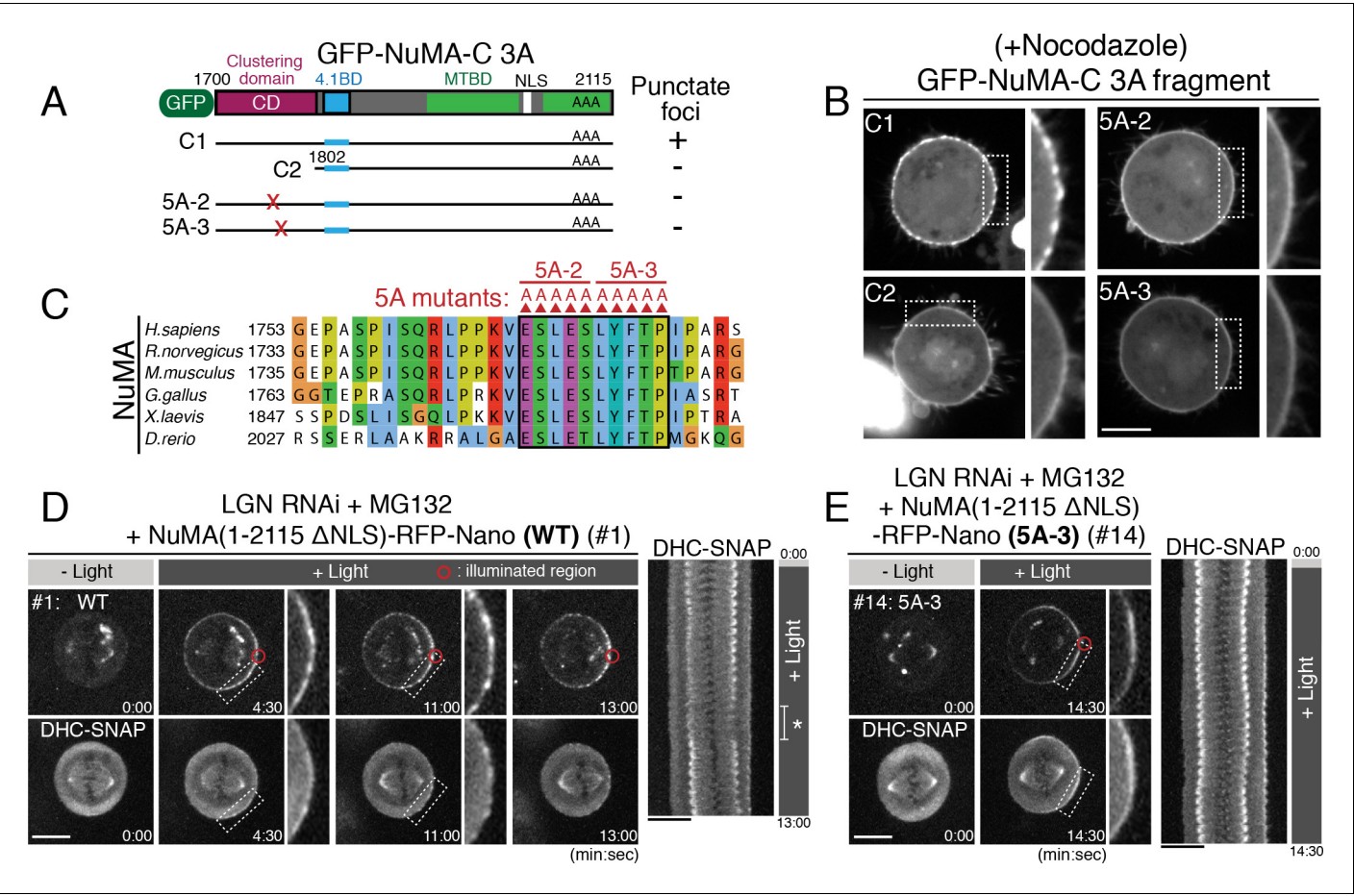

**Figure 6.** Clustering of the DDN complex by NuMA is critical for spindle pulling. (**A**) GFP-tagged NuMA C-terminal fragment and the tested NuMA mutant fragments. (**B**) Live fluorescent images of nocodazole-arrested HeLa cells expressing GFP-tagged NuMA-C 3A fragments. (**C**) Amino acid sequence alignment of the clustering domain of NuMA proteins aligned by ClustalWS. Accession numbers are indicated in *Figure 4D*. (**D–E**) Left: live fluorescent images of indicated NuMA constructs (upper) and DHC-SNAP (lower). Right: kymographs obtained from image sequences of DHC-SNAP on the left. Asterisk in (**D**) indicates the duration in which one of the spindle poles moves away from the focal plane. Scale bars = 10 μm.

DOI: https://doi.org/10.7554/eLife.36559.016

The following figure supplement is available for figure 6:

**Figure supplement 1.** Identification of a clustering domain on NuMA's C-terminal region.

DOI: https://doi.org/10.7554/eLife.36559.017

required for clustering of NuMA-C 3A, and found that the CD containing 1700–1895 region of NuMA is sufficient for NuMA fragments to form clusters at the mitotic cell cortex when targeted as a Nano fusion (*Figure 4H* and *Figure 5C*). Because this 1700–1895 fragment itself localizes to the cytoplasm, and showed no punctate signals (*Figure 6—figure supplement 1D and G* #C6), the clustering activity of this region may be enhanced by its recruitment and concentration at membranes, as observed for CRY2 clusters (*Che et al., 2015*). Consistently, NuMA's clustering function is required for spindle pulling at the cell cortex (*Figure 6E* and *Figure 7C*), but not for microtubule focusing at spindle poles (*Figure 7B*).

Interestingly, spindle pole focusing requires both NuMA's C-terminal microtubule binding and N-terminal dynein-dynactin binding modules, but not its central long coiled-coil (*Hueschen et al., 2017*). Whereas NuMA-dynein complexes generate active forces within cells, NuMA's multiple modules appear to be differently utilized depending on the context.

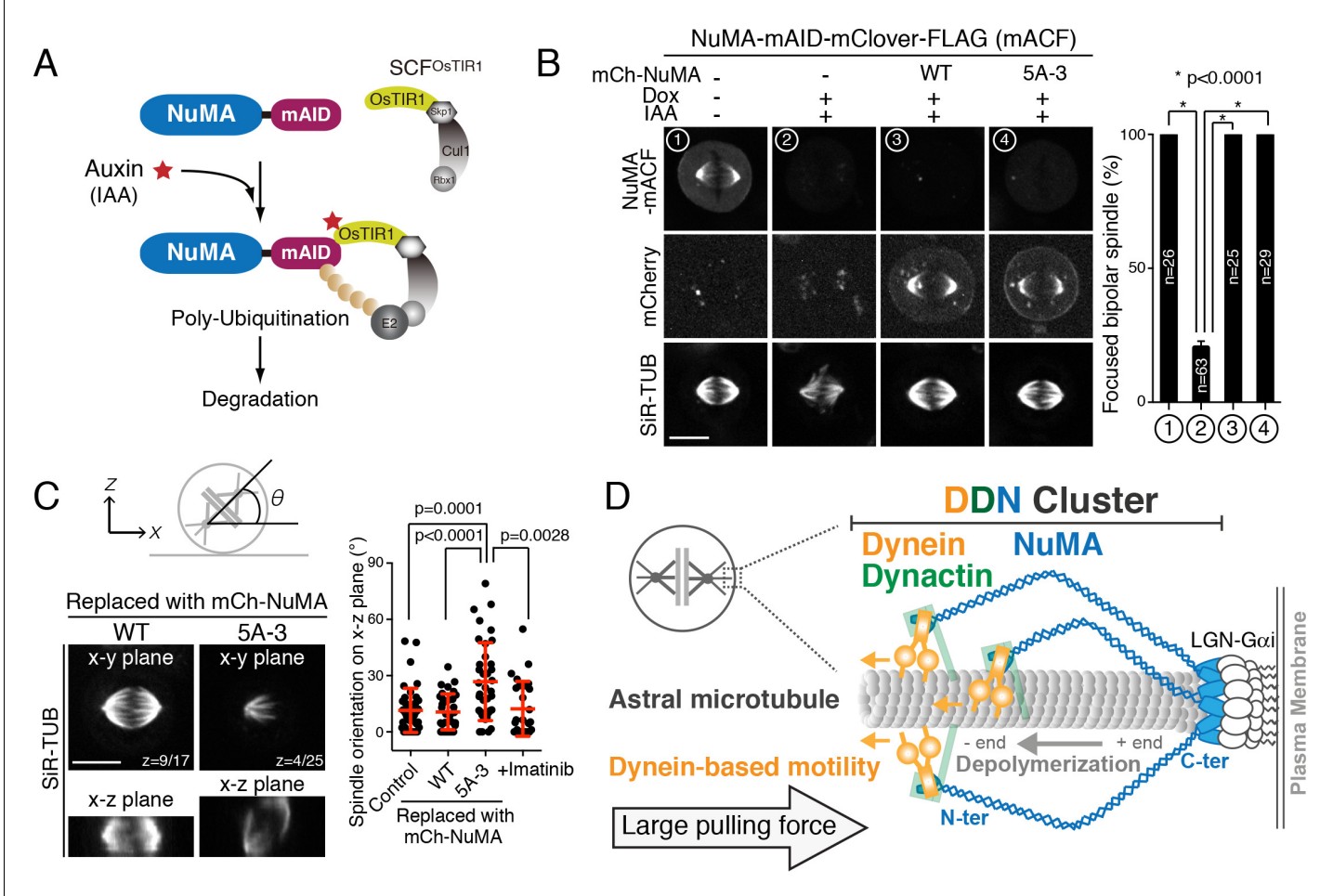

**Figure 7.** Clustering activity of NuMA is required for spindle positioning, but not for spindle pole focusing. (**A**) Diagram summarizing auxin inducible degradation (AID) system (*Natsume et al., 2016*). In the presence of OsTIR1 and auxin (IAA), mAID fusion proteins are rapidly degraded upon poly-ubiquitylation by proteasome. Because RNAi-mediated depletion of NuMA is insufficient to completely deplete NuMA proteins even after 72 hr (*Kiyomitsu and Cheeseman, 2013*), we sought to degrade endogenous NuMA using the auxin-induced degron technology. (**B**) Left: metaphase NuMA-mACF cell lines showing live fluorescent images of NuMA-mACF, mCherry-NuMA WT, or 5A3 mutant, and SiR-tubulin (SiR-TUB) after 24 hr following the treatment of Dox and IAA. The degradation of endogenous NuMA-mACF was induced by the treatment with Dox and IAA. The expression of mCherry-NuMA WT or 5A-3 was also induced by the Dox treatment. Right: histogram showing frequency of the focused bipolar spindle in each condition. * indicates statistical significance according to a Student's *t*-test (p<0.0001). Error bars indicate SEM; n > 25, from three independent experiments. (**C**) Left: orthogonal views of the metaphase spindle on the x-y (top) and x-z (bottom) plane. In each case, endogenous NuMA was replaced with either mCherry-NuMA WT or 5A-3. Right: scatterplots of the spindle orientation on the x-z plane. Red lines indicate mean ± SD; n > 27, from three independent experiments. (**D**) Model showing multiple-arm capture and pulling of an astral microtubule by the cortical DDN cluster. Scale bars = 10 μm.

DOI: https://doi.org/10.7554/eLife.36559.018

The following figure supplement is available for figure 7:

**Figure supplement 1.** Auxin-inducible degradation of endogenous NuMA and its replacement with NuMA 5A-3 mutant.

DOI: https://doi.org/10.7554/eLife.36559.019

## Mechanisms of astral-microtubule capture and pulling by the cortical DDN clusters

Our live-cell imaging revealed that DDN clusters gradually assemble at the cell cortex and then displace the spindle (*Figures 5H* and *6D*). Based on the results obtained in this study, we propose a multiple-arm capture model of astral microtubules by the DDN clusters (*Figure 7D*; *Figure 7—figure supplement 1G*). Following nuclear envelope break down, cytoplasmic NuMA and DDN complexes are recruited to the mitotic cell cortex by binding to the LGN/Gαi complex, and then

assemble DDN clusters on the cell cortex via the NuMA C-terminal domain. In vitro, up to 10–12 NuMA dimers self-assemble and form ring-like structures with an average diameter of 48 ± 8 nm (*Harborth et al., 1999*) (*Figure 7—figure supplement 1G*), which are similar to those of the central hub of yeast kinetochores (37 ± 3 nm) (*Gonen et al., 2012*), and of the Dam1 ring complex (about 50 nm) which encircles a single kinetochore microtubule (*Miranda et al., 2005*; *Westermann et al., 2005*). Given that the NuMA MTBD interacts with depolymerizing microtubules (*Seldin et al., 2016*), dynein-dynactin moves along the lattice of microtubules, and astral microtubules tends to interact with the cell cortex through an end-on configuration in pre-anaphase cells (*Kozlowski et al., 2007*; *Kwon et al., 2015*; *Samora et al., 2011*), it is tempting to speculate that the DDN cluster encircles or partially wrap the plus tip of a single astral microtubule with NuMA's MTBDs, and holds the lateral wall of the astral microtubule with multiple dynein/dynactin-containing arms (*Figure 7D*; *Figure 7—figure supplement 1G*). Future work using super-resolution imaging and in vitro reconstitution will reveal the precise architecture of the interaction between astral microtubule tips and the cortical DDN cluster. This multiple-arm capture by the DDN cluster leads to larger cooperative pulling forces by increasing the number of both dynein-dynactin containing modules and NuMA's microtubule binding per an astral microtubule. Additionally, this clustering may contribute to force generation by increasing both the stability of the DDN complex at the membrane, and the frequency for dynein-dynactin to capture or re-bind to astral microtubules. Alternatively, astral microtubule binding of the DDN complex may also assist cluster formation on the cell cortex. To produce pulling forces at dynamic plus-ends of microtubules, the cortical force-generating machinery appears to develop multiple molecular and structural features analogous to the kinetochore (*Cheeseman, 2014*; *Dimitrova et al., 2016*).

In conclusion, our optogenetic reconstitution and AID-mediated replacement reveal that the cortical DDN cluster acts as a core spindle-pulling machinery in human cells. Analyzing the structure and regulation of the DDN cluster will provide further information to understand the basis of spindle positioning in both symmetric and asymmetric cell division, and the general principles for microtubule plus-end capture and pulling.

# Materials and methods

**Key resources table**

| Reagent type (species) or resource | Designation | Source or reference | Identifiers | Additional information |
|---|---|---|---|---|
| Chemical compound, drug | SiR-tubulin | Spirochrome | Cat# SC002 | 50 nM |
| Chemical compound, drug | SiR-DNA | Spirochrome | Cat# SC007 | 20 nM |
| Chemical compound, drug | SiR-actin | Spirochrome | Cat# SC001 | 50 nM |
| Chemical compound, drug | SNAP Cell 647-SiR | New England BioLabs | Cat# S9102S | 0.1 µM |
| Chemical compound, drug | SNAP Cell TMR-star | New England BioLabs | Cat# S9105S | 0.1 µM |
| Chemical compound, drug | Hoechst 33342 | Sigma-Aldrich | Cat# B2261 | 50 ng/mL |
| Chemical compound, drug | Nocodazole | Sigma-Aldrich | Cat# M1404 | 330 nM (high dose) for 18–24 hr and 30 nM (low dose) for 1–4 hr |
| Chemical compound, drug | Paclitaxel | Sigma-Aldrich | Cat# T7402 | 10 µM |
| Chemical compound, drug | Cytochalasin D | Sigma-Aldrich | Cat# C8273 | 1 µM |
| Chemical compound, drug | MG132 | Sigma-Aldrich | Cat# C2211 | 20 µM |

*Continued on next page*

*Continued*

| Reagent type (species) or resource | Designation | Source or reference | Identifiers | Additional information |
|---|---|---|---|---|
| Chemical compound, drug | RO-3306 | Sigma-Aldrich | Cat# SML0569 | 9 µM |
| Chemical compound, drug | Imatinib mesylate | Sigma-Aldrich | Cat# SML1027 | 10 µM for 24 hr |
| Chemical compound, drug | Ciliobrevin D | Calbiochem | Cat# 250401 | 75 µM |
| Chemical compound, drug | Puromycin dihydrochloride | Wako Pure Chemical Industries | Cat# 160–23151 | 1 µg/mL |
| Chemical compound, drug | G-418 solution | Roche | Cat# 04727894001 | 800 µg/mL |
| Chemical compound, drug | Hygromycin B | Wako Pure Chemical Industries | Cat# 084–07681 | 200 µg/mL |
| Chemical compound, drug | Blasticidin S hydrochloride | Funakoshi Biotech | Cat# KK-400 | 8 µg/mL |
| Chemical compound, drug | Doxycycline hyclate | Sigma-Aldrich | Cat # D9891 | 2 µg/mL |
| Chemical compound, drug | 3-Indoleacetic acid (IAA) | Wako Pure Chemical Industries | Cat # 098–00181 | 500 µM |
| Chemical compound, drug | DirectPCR (cell) | Viagen Biotech | Cat #302 C | |
| Antibody | Anti-a-tubulin (clone DM1A) | Sigma-Aldrich | Cat# T9026 | 1:2000 |
| Antibody | Rabbit polyclonal anti-NuMA | Abcam | Cat# ab36999 (RRID:AB_776885) | 1:1000 |
| Antibody | Rabbit polyclonal anti-DHC | Santa Cruz Biotechnology | Cat# sc-9115 | 1:500 |
| Antibody | Mouse monoclonal anti-p150 | BD Transduction Laboratories | Cat# 610473 | 1:1000 |
| Antibody | Rabbit polyclonal anti-LGN | BETHYL Laboratories | Cat# A303-032A (RRID:AB_10749181) | 1:2000 |
| Antibody | Mouse monoclonal anti-Gαi-1 | Santa Cruz Biotechnology | Cat# sc-56536 | 1:100 |
| Antibody | Rabbit polyclonal anti-SNAP | New England BioLabs | Cat# P9310S | 1:1000 |
| Antibody | Rabbit polyclonal anti-OsTIR1 | Kanemaki Laboratory (*Natsume et al., 2016*) | In-house antibody | 1:1000 |
| Antibody | Rabbit polyclonal anti-phospho S10 histone H3 | Abcam | Cat# ab5176-25 | 1:500 |
| Antibody | Sheep anti-mouse IgG-HRP | GE Healthcare | Cat# NA931 | 1:10,000 |
| Antibody | Donkey anti-rabbit IgG-HRP | GE Healthcare | Cat# NA934 | 1:10,000 |
| Software, algorithm | Photoshop CS5, version 12.0 | Adobe Systems | http://www.adobe.com | |
| Software, algorithm | Fiji | (*Schindelin et al., 2012*) | https://fiji.sc/ | |
| Software, algorithm | Metamorph | Molecular Devices | https://www.moleculardevices.com | |
| Software, algorithm | GraphPad Prism 6, version 6.0 c | GraphPad Software | https://www.graphpad.com | |
| Software, algorithm | Excel | Microsoft | https://products.office.com/ | |

## Plasmid construction

Plasmids for CRISPR/Cas9-mediated genome editing were constructed according to the protocol described in Natsume et al., (*Natsume et al., 2016*). To construct CRISPR/Cas9 vectors, pX330-U6-Chimeric_BB-CBh-hSpCas9 (#42230, Addgene, Cambridge, MA) was used (*Ran et al., 2013*). PAM and 20 bp single guide RNA sequences were selected by the optimized CRISPR design tool (http://crispr.mit.edu) (*Supplementary File 2*). To construct donor plasmids containing homology arms for NuMA (~500 bp homology arms), p150 (~200 bp arms) and DHC (N-terminal,~500 bp arms), a gene synthesis service (Genewiz, South Plainsfield, NJ) was used. To construct the donor plasmid for DHC (C-terminal), a ~2,000 bp sequence was amplified by PCR from genomic DNA and then cloned into the pCR2.1-TOPO vector. A BamHI site was introduced at the center of the 2,000 bp fragment to facilitate the subsequent introduction of cassettes encoding tag and selection marker genes. To express Mem-BFP-iLID from the AAVS1 locus, membrane-targeted BFP2 ('Mem' from Neuromodulin; Clontech, Mountain View, CA) was fused to the N-terminus of iLID (#60411, Addgene) with a 53-amino acid (aa) linker derived from pIC194 (*Kiyomitsu and Cheeseman, 2012*) (#44433, Addgene), and the resulting fusion construct was introduced between the AfeI and HindIII sites in pMK231 (AAVS1 CMV-MCS-Puro, #105924, Addgene). Note that the Venus-iLID-caax construct (#60411, Addgene) was able to recruit RFP-Nano, but not NuMA-RFP-Nano to the membrane. To construct the RFP-Nano-NeoR cassette, a tagRFPt-Nano fragment (#60415, Addgene) was introduced between the SacI and MfeI sites in pMK277 (#72793, Addgene). The RFP-Nano-NeoR cassette was excised by BamHI and cloned into the BamHI site in the donor plasmid containing NuMA's homology arms. A 24-aa linker sequence containing 4 × GGGS was introduced between the last codon of NuMA and the first codon of RFP. To construct the Nano-mCherry cassette, the Nano coding sequence was fused to the N-terminal region of mCherry from pIC194 with a 2 × GGGS linker. To express Nano-mCherry-DHC, the BSDR sequence from pIC242 (*Kiyomitsu and Cheeseman, 2012*) (#44432, Addgene) was linked to the Nano-mCherry sequence with a P2A sequence, and the resulting BSDR-P2A-Nano-mCherry cassette, which contained a BamHI site at each end, was inserted into the BamHI site of the donor plasmid for DHC (N-terminal). A 47-aa linker sequence derived from pIC 194 was introduced between the last codon of mCherry and the start codon of DHC. To generate the SNAP-HygroR cassette, the mCherry coding sequence in pMK281 (#72797, Addgene) was replaced with the SNAP$_f$ coding sequence (N9186, New England BioLabs, Ipswich, MA) using In-Fusion® cloning (Takara Bio, Ōtsu, Japan). The SNAP-HygroR cassette was excised by BamHI and cloned into the BamHI site of the donor plasmids. To make the DHC donor plasmid containing a SNAP-BSDR cassette, HygroR of the SNAP-HygroR cassette was replaced with BSDR from pIC242 using In-Fusion® cloning. To conditionally express NuMA-RFP-Nano constructs from the Rosa 26 locus, a fragment containing Tet-On 3G, the TRE3GS promoter, and a multiple cloning site (MCS) derived from pMK240 (Tet-On-AAVS1-MCS-PuroR, #105925, Addgene) was introduced into pMK247 (Rosa26-CMV-MCS-HygroR, #105926, Addgene), which contains homology arms for the Rosa 26 locus. An RFP-Nano coding sequence was integrated between MluI and AgeI in the MCS, and NuMA fragments were subsequently inserted into the MluI site. NuMA truncation fragments and mutants were generated by PCR using NuMA cDNA (*Compton and Luo, 1995*; *Kiyomitsu and Cheeseman, 2012*) as a template, and the sequences were confirmed by DNA sequencing. These NuMA fragments encode isoform 2 (aa: 1–2101), which lacks a 14-aa region (aa: 1539–1552) in the longer isoform 1. However, the human NuMA constructs presented in the present study conform to isoform 1 (aa: 1–2115; NP_006176) to avoid confusion.

To construct mAID-mClover-3×FLAG NeoR, a 3 × FLAG sequence with a GGGS linker was introduced at the C-terminus of mClover of pMK289 (#72827, Addgene) by PCR. To conditionally express mCherry-NuMA WT or the 5A-3 construct from Rosa 26 locus, a fragment containing the TRE3GS promoter and the MCS derived from pMK240 was introduced into pMK247. The mCherry coding sequence derived from pIC 194 was integrated between the MluI and AgeI sites in the MCS, and the NuMA fragments were subsequently inserted Between the SalI and AgeI site.

## Cell culture and cell line generation

HCT116 and HeLa cells were cultured as described previously (*Kiyomitsu and Cheeseman, 2012*; *Natsume et al., 2016*; *Tungadi et al., 2017*). No mycoplasma contamination was detected by MycoAlert Mycoplasma Detection Kit (Lonza). Knock-in cell lines were generated according to the

procedures described in Natsume et al., (*Natsume et al., 2016*) with minor modifications. CRISPR/Cas9 and donor plasmids were transfected into the cell lines using Effectene (Qiagen, Venlo, Netherlands). For drug selection, 1 µg/mL puromycin (Wako Pure Chemical Industries, Osaka, Japan), 800 µg/mL G418 (Roche, Basel, Switzerland), 200 µg/mL hygromycin B (Wako Pure Chemical Industries), and 8 µg/mL blasticidin S hydrochloride (Funakoshi Biotech, Tokyo, Japan) were used. Selection medium was replaced with fresh selection medium 4–5 days after starting selection. After 10–14 days, colonies grown on a 10 cm culture dish were washed once with PBS, picked up with a pipette tip under a microscope (EVOS XL, Thermo Fisher Scientific, Waltham, MA) located on a clean bench, and subsequently transferred to a 96-well plate containing 50 µL of trypsin-EDTA. After a few minutes, these trypsinized cells were transferred to a 24-well plate containing 500 µL of the selection medium, and then further transferred to a 96-well plate (200 µL per well) for the preparation of genomic DNA. The remaining cells in the 24-well plate were grown and frozen using Bambanker Direct (Nippon Genetics, Tokyo, Japan). For the preparation of genomic DNA, cells in the 96-well plate were washed once with PBS and then mixed with 60 µL of DirectPCR® lysis solution (Viagen Biotech, Los Angeles, CA) containing 0.5 mg/mL proteinase K (Wako Pure Chemical Industries). The 96-well plate was sealed with an aluminum plate seal and incubated first at 56°C for 5–6 hr, then at 80°C for 2–3 hr in a water bath. To confirm the genomic insertion, PCR was performed using 1–2 µL of the genomic DNA solution and Tks Gflex DNA polymerase (Takara Bio). The cell lines and primers used in this study are listed in *Supplementary Files 1* and *3*, respectively.

Antibodies against tubulin (DM1A, Sigma-Aldrich, 1:2,000), NuMA (Abcam, 1:1,000), DHC (Santa Cruz Biotechnology, 1:500), p150 (BD Transduction Laboratories, 1:1,000), SNAP (New England BioLabs, 1:1,000), LGN (BETHYL Laboratories, 1:2,000), Gαi-1 (Santa Cruz Biotechnology, 1:100), OsTIR1 (Kanemaki Laboratory, 1:1,000), and H3S10P (Abcam, 1:500) were used for western blotting.

## Microscope system

Imaging was performed using spinning-disc confocal microscopy with a 60 × 1.40 numerical aperture objective lens (Plan Apo λ, Nikon, Tokyo, Japan). A CSU-W1 confocal unit (Yokogawa Electric Corporation, Tokyo, Japan) with three lasers (488, 561, and 640 nm, Coherent, Santa Clara, CA) and an ORCA-Flash4.0 digital CMOS camera (Hamamatsu Photonics, Hamamatsu City, Japan) were attached to an ECLIPSE Ti-E inverted microscope (Nikon) with a perfect focus system. A stage-top incubator (Tokai Hit, Fujinomiya, Japan) was used to maintain the same conditions used for cell culture (37°C and 5% $CO_2$). For light illumination, a Mosaic-3 digital mirror device (Andor Technology, Belfast, UK) and a 488 nm laser (Coherent) were used. The microscope and attached devices were controlled using Metamorph (Molecular Devices, Sunnyvale, CA).

## Immunofluorescence and live cell imaging

For immunofluorescence in *Figure 2A*, cells were fixed with PBS containing 3% paraformaldehyde and 2% sucrose for 10 min at room temperature. Fixed cells were permeabilized with 0.5% Triton X-100 for 5 min on ice, and pretreated with PBS containing 1% BSA for 10 min at room temperature after washing with PBS. Microtubules and DNA were visualized using 1:1000 anti-α-tubulin antibody (DM1A, Sigma-Aldrich, St. Louis, MO) and 1:5000 SiR-DNA (Spirochrome), respectively. Images of multiple z-sections were acquired by spinning-disc confocal microscopy using 0.2 µm spacing and camera binning 1. Maximally projected images from 15 z-sections were generated with Metamorph.

For time-lapse imaging of living cells, cells were cultured on glass-bottomed dishes (CELLview, #627870, Greiner Bio-One, Kremsmünster, Austria) and maintained in a stage-top incubator (Tokai Hit) to maintain the same conditions used for cell culture (37°C and 5% $CO_2$). Three z-section images using 0.5 µm spacing were acquired every 30 s with camera binning 2. Maximally projected z-stack images were shown in figures unless otherwise specified. Microtubules and actin were stained with 50 nM SiR-tubulin and 50 nM SiR-actin (Spirochrome), respectively, for >1 hr prior to image acquisition. DNA was stained either 20 nM SiR-DNA (Spirochrome) or 50 ng/mL Hoechst 33342 (Sigma-Aldrich) for >1 hr before observation. To visualize SNAP-tagged proteins, cells were incubated with 0.1 µM SNAP-Cell 647 SiR or TMR-STAR (New England BioLabs) for >2 hr, and those chemical probes were removed before observation.

For drug treatment, cells were incubated with drugs at the following concentrations and duration: nocodazole, 330 nM (high dose) for 18–24 hr and 30 nM (low dose) for 1–4 hr; paclitaxel, 10 µM for

1–10 min; cytochalasin D, 1 µM for 1–10 min; MG132, 20 µM for 1–4 hr (*Figure 4—figure supplement 1B*); RO-3306, 9 µM for 20 hr; imatinib, 10 µM for 24 hr (*Matsumura et al., 2012*); doxycycline hyclate (Dox), 2 µg/mL (*Figure 4—figure supplement 1B*); Ciliobrevin D, 75 µM.

To express NuMA-RFP-Nano constructs from the Rosa 26 locus in LGN-depleted cells, cells were treated with LGN siRNA (*Kiyomitsu and Cheeseman, 2012*) and Dox at 24 hr and 48 hr, respectively, according to the procedure described in *Figure 4—figure supplement 1B*. RO-3306 was added at 48 hr to cells that were then synchronized at G2 at 68 hr. The NuMA-RFP-Nano fusion protein was expressed in most cells, but its expression frequency was reduced in cells that expressed longer NuMA fragments. siRNAs targeting Gαi-1 isoforms (*Kiyomitsu and Cheeseman, 2012*) were obtained from Dhamacon.

To compare the intensities of cortically targeted NuMA-Nano fusions, images of NuMA-Nano fusions and DHC-SNAP were acquired using the same parameters (Exposure time: NuMA, 1000 msec; DHC, 500 msec), except for *Figure 1B* (NuMA, 1500 msec; DHC, 500 msec). To optimize image brightness, same linear adjustments were applied using Fiji and Photoshop. Supplemental movie files were generated using Metamorph and Fiji.

To activate the auxin-inducible degradation of NuMA-mAID-mClover-3FLAG (mACF), cells were treated with 2 µg/mL Dox and 500 µM indoleacetic acid (IAA) for 20–24 hr. Cells with undetectable mClover signals were analyzed. A small population of cells showed mClover signals even after being treated with Dox and IAA. For replacement experiments, either mCherry-NuMA WT or the 5A-3 mutant was expressed from the Rosa 26 locus following Dox treatment. This caused the cells to simultaneously express OsTIR1 from the AAVS1 locus to initiate the auxin-inducible degradation of endogenous NuMA-mACF.

## Light-inducible targeting

Except for *Figure 1—figure supplement 1B*, HCT116 cells expressing Mem-BFP-iLID and NuMA-Nano fusion proteins were treated with RO-3306 and MG-132 according to the procedure described in *Figure 4—figure supplement 1B* to increase the proportion of metaphase-arrested cells.

To target Nano fusion proteins at the metaphase cell cortex, cells were illuminated using a Mosaic-3 digital mirror device (Andor Technology) at the indicated regions (circles with a diameter of 1.95 µm for *Figure 1—figure supplement 1B*, and that of 2.82 µm for other figures) with a 488 nm laser pulse (500 msec exposure, 25 mW). To manually control the frequency of the light pulse and the position of the illuminated region during time-lapse experiments, a custom macro was developed using Metamorph. Using this macro, indicated regions were illuminated ~10 times with the light pulse during time intervals (30 s) between image acquisitions. The illuminated position was adjusted to precisely illuminate the cortical region of each cell. In response to the expression level of the Nano fusion proteins, the frequency of the light pulse was reduced to prevent the targeting of Nano fusion proteins throughout the cell cortex.

To reposition NuMA-RFP-Nano at the mitotic cell cortex in *Figure 1F,* a cortical region adjacent to the spindle axis was illuminated. The light-illuminated region was changed once the spindle started to move but before the spindle was completely attached to the cell cortex. Spindles that rotated by approximately 90° within 15 min were counted.

## Quantification of cortical fluorescent signals and spindle displacement

Cortical and cytoplasmic fluorescence intensities were determined using Fiji by calculating the mean pixel intensity along three different straight lines (length 3 µm, width three pixels) drawn along the cell cortex showing Nano signals or the cytoplasm near the cell cortex but without any aggregations. The background intensity was subtracted from each measurement. The distance from the pole to the cell cortex was measured using Metamorph or Fiji. Line scans for cortical fluorescence intensity were generated using Fiji by calculating the mean pixel intensity along the segmented line (width three pixels) drawn along the cell cortex. Kymographs were generated using Photoshop (Adobe Systems, San Jose, CA).

Spindle displacement was judged by the definition given in *Figure 4—figure supplement 1I*. In addition, cells that satisfied the following conditions were analyzed; (1) NuMA-RFP-Nano fusion proteins were asymmetrically recruited at the light-illuminated region, but not distributed to a whole cell cortex. (2) The cortical intensities of NuMA-Nano fusion proteins were higher than that of NuMA

Δex24-RFP-Nano (*Figure 5F*). (3) DHC-SNAP was detectable at the light-illuminated region except for the case of the cortical targeting of NuMA-C (#13). (4) The spindle was monitored for >10 min, and not vertically rotated. (5) The bipolar spindle was properly formed without severe membrane blebbing.

## Statistical analysis

To determine the significance of differences between the mean values obtained for two experimental conditions, Student's *t*-tests or Mann-Whitney tests (Prism 6; GraphPad Software, La Jolla, CA) were used as indicated in the figure legends.

## Acknowledgements

We thank I M Cheeseman and G Goshima for advice and critical reading of the manuscript, R Inaba, M Nishina, K Murase, and Y Tsukada for technical assistance, T Nishiyama and A Sasaki for reagents, and PRESTO members for discussion. This work was supported by grants from PRESTO program (JPMJPR13A3) of the Japan Science and Technology agency (JST), a Career Development Award of the Human Frontier Science Program (CDA00057/2014 C), KAKENHI (16K14721, 17H05002) of the Japan Society for Promotion of Science (JSPS), Collaborative Research Program (2014-B, 2015-A1, 2016-A1) of the National Institute of Genetics (NIG), the Uehara Memorial Foundation, the Nakajima Foundation, and the Naito Foundation.

## Additional information

### Funding

| Funder | Grant reference number | Author |
|---|---|---|
| Japan Science and Technology Agency | JPMJPR13A3 | Tomomi Kiyomitsu |
| Human Frontier Science Program | CDA00057/2014-C | Tomomi Kiyomitsu |
| Japan Society for the Promotion of Science | 16K14721 | Tomomi Kiyomitsu |
| Uehara Memorial Foundation | | Tomomi Kiyomitsu |
| Naito Foundation | | Tomomi Kiyomitsu |
| Japan Society for the Promotion of Science | 17H05002 | Tomomi Kiyomitsu |
| The Nakajima Foundation | | Tomomi Kiyomitsu |
| National institute of Genetics | Collaborative Research Program (2014-B) | Tomomi Kiyomitsu |
| National Institute of Genetics | Collaborative Research Program (2015-A-1) | Tomomi Kiyomitsu |
| National Institute of Genetics | Collaborative Research Program (2016-A1) | Tomomi Kiyomitsu |

The funders had no role in study design, data collection and interpretation, or the decision to submit the work for publication.

### Author contributions

Masako Okumura, Data curation, Investigation; Toyoaki Natsume, Masato T Kanemaki, Methodology; Tomomi Kiyomitsu, Conceptualization, Resources, Data curation, Software, Formal analysis, Supervision, Funding acquisition, Validation, Investigation, Visualization, Methodology, Writing—original draft, Project administration, Writing—review and editing

### Author ORCIDs

Toyoaki Natsume (iD) http://orcid.org/0000-0002-3544-4491
Masato T Kanemaki (iD) http://orcid.org/0000-0002-7657-1649
Tomomi Kiyomitsu (iD) http://orcid.org/0000-0002-2280-4611

**Decision letter and Author response**
Decision letter https://doi.org/10.7554/eLife.36559.026
Author response https://doi.org/10.7554/eLife.36559.027

## Additional files

**Supplementary files**
• Supplementary file 1. Cell lines used in this study.
DOI: https://doi.org/10.7554/eLife.36559.020

• Supplementary file 2. sgRNA sequences for CRISPR/Cas9-mediated genome editing.
DOI: https://doi.org/10.7554/eLife.36559.021

• Supplementary file 3. PCR primers to confirm gene editing.
DOI: https://doi.org/10.7554/eLife.36559.022

• Transparent reporting form
DOI: https://doi.org/10.7554/eLife.36559.023

**Data availability**

All data generated or analyzed during this study are included in the manuscript, figures and supplemental files. We will deposit all plasmids and cell lines used in this study to non-profit organization such as Addgene and RIKEN BioResource Research Center.

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
