## [Decision Letter]

Thank you for submitting your article "Dynein-Dynactin-NuMA clusters generate cortical spindle-pulling forces as a multi-arm ensemble" for consideration by *eLife*. Your article has been reviewed by three peer reviewers, and the evaluation has been overseen by a Reviewing Editor and Andrea Musacchio as the Senior Editor. The reviewers have opted to remain anonymous.

The reviewers have discussed the reviews with one another and the Reviewing Editor has drafted this decision to help you prepare a revised submission.

Summary:

All three reviewers were strongly supportive of your manuscript. You use of an elegant optogenetics approach to systematically dissect the molecular requirements for dynein-mediated cortical pulling on astral microtubules, which determines the position of mitotic spindles. Direct light-induced targeting of NuMA to the cortex bypasses the requirement for NuMA's cortical receptor LGN and allows the precise definition of the domains and motifs in NuMA required for dynein-dynactin recruitment and cortical force generation. The generation of the innovative experimental toolbox is carefully documented, and the technical standard of the experiments is high.

Using these tools, with proteins at endogenous levels, you dynamically control spindle position and orientation. You determine that astral microtubule depolymerization is not required for pulling, and test the function of a large panel of NuMA truncations and mutants. You also identify a motif in NuMA's N-terminus that is required for cortical dynein recruitment, and show that a motif in NuMA's C-terminus is required for the punctate foci of NuMA and dynein seen at the cell cortex. Mutations that prevent the formation of punctate foci also prevent cortical pulling, so you conclude that NuMA-mediated clustering of dynein-dynactin-NuMA is required for cortical pulling and spindle positioning. This is a conceptually novel point and constitutes the main advance of the paper.

In addition, this work opens questions on NuMA's multimeric organization (e.g., the regulation and function of oligomerization at the cortex vs. within the spindle), and it provides insight into the big picture question of how active forces within cells are spatially organized. The work is characterized by careful experimental design and thorough controls, and the manuscript and figures are clear and well presented (with helpful schematics and organization).

Essential revisions:

Experimental:

The reviewers ask for two pieces of experimental data. The first needs to be addressed if possible. The second would clearly add to the paper, but the reviewer leaves it up to you as to whether you want to include it – if not then you should address the reviewers questions.

1) Most of the experiments presented are conducted in an LGN silenced background, thus it seems that pulling forces do not require LGN nor Gαi. For the sake of completeness, it would be beneficial to show LGN levels at the end of the experiments, i.e. after 72 hours from siRNA treatment, and at the same time test the effect of removing Gαi.

2) Dynein-dynactin recruitment requires a Spindly-like motif in NuMA's N-terminal region. This is an intriguing finding that suggests mechanistic similarities between functionally diverse dynein adaptors. However, the authors also show that NuMA's N-terminal region is not sufficient for cortical force generation. This raises the question of how good NuMA's N-terminal region really is at activating dynein. For comparison, it might be interesting to check whether cortical targeting of the N-terminal BICD2 region, which robustly activates dynein-dynactin motility in vitro and in vivo, is sufficient for force generation in this assay.

Manuscript revisions:

The reviewers have also identified a number of points that need clarification in order to improve the manuscript. I attach the list of these below. Please address these points in your response to the reviewers and make changes to the manuscript where appropriate.

Reviewer #1:

- Robust spindle displacement is observed after addition of taxol (Figure 2). This is surprising given the end-on microtubule binding by NuMA clusters proposed by authors, which requires shortening of microtubules to pull the spindle towards the cortex. The authors should comment on how the taxol result is compatible with their model.

- The authors show that direct targeting of dynein to the cortex (which also results in dynactin recruitment) is not sufficient per se for the generation of cortical pulling forces and that cortical force generation requires the second microtubule binding domain of NuMA. These findings are in agreement with the results of Seldin, Muroyama, and Lechler (2016), who showed that the microtubule-binding activity of NuMA is required for mitotic spindle positioning in cultured mouse keratinocytes and the epidermis. This should be explicitly acknowledged by the authors.

- The generation of cortical pulling forces requires the central coiled-coil region of NuMA. This contrasts with the observation that the same NuMA region is dispensable for spindle pole focusing, as shown by Hueschen et al. (*eLife* 2017). The context-dependent requirement for NuMA's central coiled-coil region is interesting and should be commented on.

- The generation of cortical pulling forces requires clustering of NuMA at the cortex, which is mediated by a highly conserved patch in NuMA's C-terminal region. This is perhaps the most important conclusion of this study and suggests a mechanism similar to the one used by the kinetochore, where multiple microtubule-binding activities are concentrated for stable end-on attachment. The authors should clarify the following points: does NuMA clustering occur independently of astral microtubules? In the model (Figure 7D), LGN clusters with NuMA – but is there evidence for this?

- Figure 4C: the number of cells examined in the 'spindle displacement' assay for NuMA fragments #3 and #6 is too low (2 and 3, respectively).

Reviewer #2:

- The light-induced system adopted in the article enables local recruitment of NuMA-RFP-Nano (or Dynein-RFP-Nano) to the cortex by hetero-dimerization of the fusion proteins with activated membrane-bound iLID. It is known that NuMA cortical levels are critical in determining the extent of spindle pulling forces. What are the levels of NuMA-RFP-Nano and Dynein-RFP-Nano recruited at the cortex upon light induction compared to the levels of endogenous cortical NuMA and Dynein in metaphase? Also, in Figure 1B-F, the authors show that illuminating a tiny portion of the cortex is sufficient to recruit extended cortical crescents of NuMA. How do the authors explain this extended NuMA cortical distribution molecularly?

- In Figure 4, the authors engineer RFP-Nano NuMA fragments and test their ability to recruit Dynein at the cortex and position the spindle. As a positive control they use NuMA-Δ-NLS explaining that it does not dimerize with endogenous NuMA. Can the authors explain why NLS removal would prevent dimerisation?

- A major finding of the manuscript is the identification of cortical NuMA/Dynein clusters that are required for astral MT-pulling forces. A critical issue is to which extent the clustering might be affected by the protein engineering used to visualise and localise the proteins.

In the presented experiments, the levels of NuMA at the cortex seem to vary significantly depending on the construct (see Figure 4 and Figure 5). In addition, in the Figure 5—figure supplement 1 the authors report that the construct 1-1985 tends to aggregate at the cortex. Can the authors comment on the specificity of the dotted pattern they see – i.e. whether the constructs are stable and soluble? One consideration that comes to my mind is that the 5A-2 and 5A-3 residue stretches of NuMA that are responsible for clustering correspond to hydrophobic residues, that might be engaged in intramolecular interactions maintaining the protein stability. Of note, for some reasons, the dotted pattern does not seem evident in Figure 1.

On the same line of thoughts, in Figure 6B and Figure 6—figure supplement 1C, the authors use GFP-NuMA-C-3A mutant that localizes at the cortex as it lacks the inhibitory cdk1 phosphorylation sites. In a wild-type background, this construct should be recruited at the membrane by LGN. Is the dotted structure of GFP-NuMA-C visible upon LGN cortical recruitment? In Figure 6—figure supplement 1C the authors show that the punctuated NuMA-3A pattern intercalates with the cortical actin pattern. This is an interesting observation, can the authors comment on this?

- One of the issues connected to the idea of NuMA clusters relates to the molecular events underlying their formation under physiological conditions. The authors identify NuMA residues responsible for the clustering: do they think these residues mediate self-oligomerization of NuMA? In the experiments of Figure 7B and Figure 7—figure supplement 1F, in which endogenous NuMA is replaced with full-length NuMA either wild-type or 5A-3, the cortical clustering does not seem very evident.

- In the Discussion, the authors propose that NuMA clustering works within a DDN network to promote end-on attachment of astral microtubules to the cortex. To my knowledge, no compelling evidence for end-on attachment in this process has been provided. This model, although possible, tends to reflect what known for the attachment of microtubules to kinetochore. Therefore, in the absence of specific evidence, it might be preferable to down-state the model.

Reviewer #3:

1) The idea that NuMA may activate dynein in addition to regulating its localization is an attractive hypothesis, which has been proposed by others but not carefully tested. Similarly, this work includes no direct data related to dynein activation. The authors show that the Spindly-like motif is required for dynein recruitment, but they do not decouple recruitment (localization) and activation. Thus, the use of language throughout the paper referring to dynein activation could be misleading; it should be softened and mostly confined to the Discussion. It should be made clear that the authors did not test whether NuMA activates dynein.

2) In the experiments in Figures 4 and 5, why is NuMA 1-2115 ΔNLS used instead of the full-length protein? In the subsection “A Spindly-like motif in NuMA is required for cortical dynein recruitment, but not sufficient for spindle pulling”, the authors state that the NLS is removed to prevent dimerization with endogenous protein, but do not state a source. Do they believe this to be true, and due to what published or unpublished data? Published work (e.g., Harborth et al., EMBO 1995) suggests that dimerization will still occur through NuMA's coiled-coils.

3) Relatedly, dimerization with endogenous NuMA presumably occurs with NuMA truncations in Figures 4 and 5 that include some or all of the central coiled-coil. Could the authors comment on how this dimerization could affect their conclusions based on these data? As one example, could the inability of fragment #12 reflect an inability to form a functional dimer with endogenous protein due to absence of big part of the central coiled-coil?

4) In Figure 4C and 5B, the application of a z-test to "yes/no" data (not normally distributed) with low "N" is likely inappropriate. We suggest using the actual pole displacement distances (as a percentage of starting pole-to-cortex distance) for each condition, and then analyzing those data using an ANOVA (or, at least, a t-test between the control (#1) and each condition). Consulting a statistician may be helpful.

5) Relatedly, directly reporting the actual pole displacement distances for NuMA fragments #1-14 (Figure 4-5) would be interesting and important for the reader to interpret the data. These data could be displayed as "beeswarm" plots like Figure 7C, for example, and they would help the reader better interpret the significance of your findings (for example, it would better display the degree of spindle displacement difference between NuMA fragment #11 (3/8) and fragment #14 (1/13)).

6) In model Figure 7D (and related Figure 7—figure supplement 1), the main features of the cartoon should stay as close to what was shown in the paper as possible. If there is key speculation not supported by data, it should be made clear that it is speculation. In particular, whether LGN forms a ring does not seem to be based on data and drawing LGN and NuMA's C-terminus as more disorganized may avoid readers being biased towards a ring-like model (alternatively, if a ring-like model is supported by data, it should be made clear what data supports it).

---

## [Author Response]

Essential revisions:Experimental:The reviewers ask for two pieces of experimental data. The first needs to be addressed if possible. The second would clearly add to the paper, but the reviewer leaves it up to you as to whether you want to include it – if not then you should address the reviewers questions.1) Most of the experiments presented are conducted in an LGN silenced background, thus it seems that pulling forces do not require LGN nor Gαi. For the sake of completeness, it would be beneficial to show LGN levels at the end of the experiments, i.e. after 72 hours from siRNA treatment, and at the same time test the effect of removing Gαi.

We performed a Western blot to demonstrate the reduction of LGN protein level after siRNA treatment. This data is now included in Figure 1—figure supplement 1H. In addition, we conducted light-induced NuMA targeting in a Gαi (1+2+3) silenced background. The metaphase spindle was displaced toward the light-illuminated region in 71.4% of cells (n=7), similarly to in LGN-depleted cells. These data are now included in Figure 1—figure supplement 2E-F. Taken together, these results indicate that cortical pulling forces do not require LGN or Gαi in our optogenetic system.

2) Dynein-dynactin recruitment requires a Spindly-like motif in NuMA's N-terminal region. This is an intriguing finding that suggests mechanistic similarities between functionally diverse dynein adaptors. However, the authors also show that NuMA's N-terminal region is not sufficient for cortical force generation. This raises the question of how good NuMA's N-terminal region really is at activating dynein. For comparison, it might be interesting to check whether cortical targeting of the N-terminal BICD2 region, which robustly activates dynein-dynactin motility in vitro and in vivo, is sufficient for force generation in this assay.

We have now cloned human N-terminal BICD2 region (1-400 aa) and generated cell line that conditionally expresses BICD2-N-RFP-Nano in the presence of Dox. Unexpectedly, light-induced targeting of the BICD2-N construct was not sufficient to recruit dynein (DHC-SNAP) to the mitotic cell cortex (n=10), although the BICD2-N construct was able to partially recruit dynein to the plasma membrane in interphase (n=5/8 cells) (Please see Author response image 1). These results suggest that interaction between dynein and BICD2-N is during cell cycle, and thus BICD2-N targeting is not a feasible way to dissect dynein-based force generation during mitosis. Because of these reasons, we did not include these data in the revised manuscript.

Alternatively, to understand whether light-induced NuMA-dynein complexes require dynein activity for spindle displacement, we analyzed the effect of ciliobrevin D on force generation. This drug inhibits dynein-dependent microtubule gliding and ATPase activity, but not the association between ADP-bound dynein and microtubules in vitro (Firestone et al., 2012). In HCT116 cells, we found that ciliobrevin D treatment in interphase caused mitotic phenotypes including chromosome misalignment similar to dynein degradation (Natsume et al., 2016) under 0.5%, but not 10% FBS culture conditions (Figure 3—figure supplement 1B-D), consistent with previous reports (Firestone et al., 2012). We next added ciliobrevin D to metaphase-arrested cells. Although dynein activity is required to maintain spindle bipolarity, we found that spindle bipolarity was maintained for ~30 min following the treatment of ciliobrevin D, and was gradually disrupted during the subsequent 30-60 min (Figure 3—figure supplement 1E-G). Thus, we next sought to perform spindle pulling assay during the initial 60 min according to the Procedure depicted in Figure 3A. In control cells, light-induced targeting of NuMA displaced the spindle in 80% of cell (n=10, Figure 3B and D). In contrast, the spindle was not displaced in 75% of ciliobrevin D-treated cells (n=12, Figure 3C-D), whereas dynein was normally recruited to the cell cortex and the spindle structure was maintained during the assay. These results suggest that light-induced NuMA activates dynein and its activity is required for generating cortical pulling forces. These data are now included in Figure 3A-D, and Figure 3—figure supplement 1A-G.

We also performed the same assay using cells expressing NuMA’s N-terminal fragment (1-705 aa; Figure 4B, C #3) to test whether asymmetric spindle-pole enrichment of the NuMA N-terminal fragment (Figure 4F) depends on dynein activity. However, in 0.5% FBS culture condition, the N-terminal NuMA fragment displays large cytoplasmic aggregation, which accumulates around spindle poles during mitosis, and prevented us from analyzing asymmetric spindle-pole enrichment of this fragment after light-induced cortical targeting.

In total, our results indicate that NuMA not only recruits dynein-dynactin to the mitotic cell cortex but also activates dynein at the cell cortex to generate cortical pulling forces. Although it is still unclear how good NuMA’s N-terminal region activates dynein, we believe that these new additions provide a strong evidence that dynein is activated following NuMA-mediated cortical recruitment to generate functional spindle-pulling forces.

Manuscript revisions:The reviewers have also identified a number of points that need clarification in order to improve the manuscript. I attach the list of these below. Please address these points in your response to the reviewers and make changes to the manuscript where appropriate.Reviewer #1:- Robust spindle displacement is observed after addition of taxol (Figure 2). This is surprising given the end-on microtubule binding by NuMA clusters proposed by authors, which requires shortening of microtubules to pull the spindle towards the cortex. The authors should comment on how the taxol result is compatible with their model.

In taxol-treated cells, the velocity of the spindle movement was slower than that observed in control cells (Figure 2I), suggesting that depolymerization of astral microtubules may also contribute to force generation as described in the model. Of course, this reduced velocity may be due to other reasons such as cortical pushing by stabilized astral microtubules. Thus, we have added comments to the revised manuscript as described below.

“In these taxol-treated cells, the velocity of the spindle movement was slower than that observed in control cells (Figure 2F-I), suggesting that depolymerization of astral microtubules may also contribute to force generation, although the reduced velocity might be caused alternatively by cortical pushing by stabilized astral microtubules.”

- The authors show that direct targeting of dynein to the cortex (which also results in dynactin recruitment) is not sufficient per se for the generation of cortical pulling forces and that cortical force generation requires the second microtubule binding domain of NuMA. These findings are in agreement with the results of Seldin, Muroyama, and Lechler (2016), who showed that the microtubule-binding activity of NuMA is required for mitotic spindle positioning in cultured mouse keratinocytes and the epidermis. This should be explicitly acknowledged by the authors.

We now cited this paper as follows:

“… direct binding of NuMA to astral microtubules may generate cooperative forces in parallel with dynein-dynactin recruitment as recently proposed by Seldin et al. (Seldin, Muroyama, and Lechler, 2016).”

In addition, we have added comments for why NuMA Δex24, which corresponds to the mouse NuMAΔex22 mutant used in Seldin et al. paper, is still be able to generate pulling forces in our assay as follows.

“Because the corresponding mouse NuMA Δex22 mutant shows spindle orientation defect in mouse keratinocytes and the epidermis (Seldin et al., 2016), this region may have specific roles in different cell types. Alternatively, weak defects in the NuMA Δex24 mutant may be suppressed by targeting increased levels of cortical NuMA Δex24 in this assay.”

- The generation of cortical pulling forces requires the central coiled-coil region of NuMA. This contrasts with the observation that the same NuMA region is dispensable for spindle pole focusing, as shown by Hueschen et al. (2017). The context-dependent requirement for NuMA's central coiled-coil region is interesting and should be commented on.

We appreciate this comment. The following sentences are now included in Discussion:

“Interestingly, spindle pole focusing requires both NuMA’s C-terminal minus-end binding and N-terminal dynein-dynactin binding modules, but not its central long coiled-coil (Hueschen et al., 2017). Whereas NuMA-dynein complexes generate active forces within cells, NuMA’s multiple modules appear to be differently utilized depending on the context.”

- The generation of cortical pulling forces requires clustering of NuMA at the cortex, which is mediated by a highly conserved patch in NuMA's C-terminal region. This is perhaps the most important conclusion of this study and suggests a mechanism similar to the one used by the kinetochore, where multiple microtubule-binding activities are concentrated for stable end-on attachment. The authors should clarify the following points: does NuMA clustering occur independently of astral microtubules?

The GFP-NuMA-C 3A fragment shows punctate signals in nocodazole-arrested cells (Figure 6B). In addition, NuMA forms oligomers in vitro (Harborth et al., 1999). Thus, we believe that NuMA is able to cluster independently of astral microtubules. We have now added (+Nocodazole) in Figure 6B. However, we cannot exclude the possibility that astral microtubules assist NuMA clustering in cells. We have now included the following sentence in the Discussion:

“Alternatively, astral microtubule binding of the DDN complex may also assist cluster formation on the cell cortex.”

In the model (Figure 7D), LGN clusters with NuMA – but is there evidence for this?

We generated a NuMA-mACF and SNAP-LGN double knock-in HCT116 cell line, and visualized both NuMA and LGN. We found that LGN also displays punctate cortical signals as observed in Rpe1 cells (Figure S4b in Kiyomitsu and Cheeseman, 2012). We now included this data in Figure 7—figure supplement 1C.

- Figure 4C: the number of cells examined in the 'spindle displacement' assay for NuMA fragments #3 and #6 is too low (2 and 3, respectively).

We have now repeated this experiment using the NuMA #3 cell line, and increased the number of cells up to 7 in the spindle displacement assay. Together with the results of NuMA #2 construct (n=14), these data indicate that NuMA’s N-terminal fragment is not sufficient for spindle displacement. For the NuMA #6 cell line, we have changed this to n.d. (not determined) in Figure 4C because of the low number of cells.

Reviewer #2:- The light-induced system adopted in the article enables local recruitment of NuMA-RFP-Nano (or Dynein-RFP-Nano) to the cortex by hetero-dimerization of the fusion proteins with activated membrane-bound iLID. It is known that NuMA cortical levels are critical in determining the extent of spindle pulling forces. What are the levels of NuMA-RFP-Nano and Dynein-RFP-Nano recruited at the cortex upon light induction compared to the levels of endogenous cortical NuMA and Dynein in metaphase?

Although we have quantified the level of NuMA-RFP-Nano and DHC-SNAP in the previous Figure 1—figure supplement 1, we did not clearly describe these data. We now describe this data as follows.

“The level of light-induced cortical NuMA is about 3 times higher than that of endogenous NuMA in metaphase, but similar to that in anaphase (Figure 1—figure supplement 1I-J).”

The level of cortical Nano-mCherry-DHC is also quantified in Figure 3G. This Nano-mCherry- DHC level is higher than that of endogenous SNAP-DHC in Figure 1C, suggesting that failure of force generation by light-induced dynein targeting is not due to its lower level of cortical dynein.

Also, in Figure 1B-F, the authors show that illuminating a tiny portion of the cortex is sufficient to recruit extended cortical crescents of NuMA. How do the authors explain this extended NuMA cortical distribution molecularly?

One reason for the illumination of a small region promoting crescent formation is the diffusion of NuMA-RFP-Nano/ Mem-BFP-iLID complex on the plasma membrane. These complexes can move about 5 μm within 60 sec when a diffusion constant (D = 0.1 μm^2^/sec) for a typical membrane bound protein is used for the calculation. In addition, this illumination may be affected by technical reasons related to the optics. In principle, the laser is focused and illuminated at the defined region on the focal plane with maximum power. However, the laser also penetrates its vertical region with wider breadth. Because mitotic cells have a round shape compared to interphase cells, such wider vertical illumination may stimulate its vertical region, and cause wider distribution of NuMA-RFP-Nano on the focal plane following diffusion. Other reasons such as light scattering by unknown cellular components may also cause a wider illumination. Nonetheless, endogenous NuMA and dynein normally extend along the cell cortex as shown in Figure 1B left. Thus, we believe that the extended NuMA cortical distribution is similar to its native localization and is appropriate to assess its ability for cortical force generation.

- In Figure 4, the authors engineer RFP-Nano NuMA fragments and test their ability to recruit Dynein at the cortex and position the spindle. As a positive control they use NuMA-Δ-NLS explaining that it does not dimerize with endogenous NuMA. Can the authors explain why NLS removal would prevent dimerisation?

We removed the NLS to spatially separate exogenously expressed NuMA constructs from endogenous NuMA, which dominantly localizes to the nucleus in interphase cells. In fact, based on our synchronized expression protocol as described in Figure 4—figure supplement 1B, exogenously expressed NuMA constructs localized to the cytoplasm before G2 release (at 68 hr), and separated from endogenous NuMA in the nucleus, suggesting that the large majority of exogenously expressed NuMA mutants form homo-dimers through its coiled-coil in the cytoplasm, but not with endogenous NuMA. The following sentences are now included:

“the NLS was deleted to reduce dimerization with endogenous NuMA by spatially separating exogenously expressed constructs from nuclear-localized endogenous NuMA before G2 release.”

- A major finding of the manuscript is the identification of cortical NuMA/Dynein clusters that are required for astral MT-pulling forces. A critical issue is to which extent the clustering might be affected by the protein engineering used to visualise and localise the proteins.In the presented experiments, the levels of NuMA at the cortex seem to vary significantly depending on the construct (see Figure 4 and Figure 5). In addition, in the Figure 5—figure supplement 1 the authors report that the construct 1-1985 tends to aggregate at the cortex. Can the authors comment on the specificity of the dotted pattern they see – i.e. whether the constructs are stable and soluble? One consideration that comes to my mind is that the 5A-2 and 5A-3 residue stretches of NuMA that are responsible for clustering correspond to hydrophobic residues, that might be engaged in intramolecular interactions maintaining the protein stability. Of note, for some reasons, the dotted pattern does not seem evident in Figure 1.

At present, we do not have data regarding the stability and solubility of each NuMA construct. To answer this question, future work will be necessary to express these constructs in vitro and analyze their structural and biochemical properties, which we believe is beyond the scope of this study. Regarding the dotted pattern, smaller constructs such as NuMA-C show more visible punctate signals. Although we repeatedly observe punctate signals for full length NuMA, it is difficult to recognize clear dots as for NuMA-C. This may indicate dynamic assembly and/or reorganization of NuMA clusters in addition to their lateral diffusion on the cell cortex. Alternatively, NuMA-C may form additional oligomeric structure. We have now included the following sentence in the Results section.

“… we found that NuMA constructs containing its C-terminal region displayed punctate cortical signals, which tend to be more evident in smaller constructs (e.g. Figure 5H-I)”.

On the same line of thoughts, in Figure 6B and Figure 6—figure supplement 1C, the authors use GFP-NuMA-C-3A mutant that localizes at the cortex as it lacks the inhibitory cdk1 phosphorylation sites. In a wild-type background, this construct should be recruited at the membrane by LGN. Is the dotted structure of GFP-NuMA-C visible upon LGN cortical recruitment?

Although NuMA-C contains the LGN binding domain, GFP-NuMA-C WT is hardly detectable at the metaphase cell cortex (Please see Figure 1F and 1G in Kiyomitsu and Cheeseman, 2013), suggesting that NuMA-C is not sufficient to interact with LGN. In fact, LGN was not detected as a GFP-NuMA-C 3A binding proteins in our previous mass spectrometry analysis (Kiyomitsu and Cheeseman, 2013), and is dispensable for punctate signal formation of GFP-NuMA-C-3A (Figure 2C in Kiyomitsu and Cheeseman, 2013).

In Figure 6—figure supplement 1C the authors show that the punctuated NuMA-3A pattern intercalates with the cortical actin pattern. This is an interesting observation, can the authors comment on this?

We appreciate this comment. We have now added following comments to the text:

“Interestingly, the punctate NuMA-C 3A patterns intercalated with cortical actin localization, and still existed following actin disruption (Figure 6—figure supplement 1C). These results suggest that the NuMA C-terminal fragment self-assembles on the membrane independently of its cortical binding partners and actin cytoskeleton.”

- One of the issues connected to the idea of NuMA clusters relates to the molecular events underlying their formation under physiological conditions. The authors identify NuMA residues responsible for the clustering: do they think these residues mediate self-oligomerization of NuMA? In the experiments of Figure 7B and Figure 7—figure supplement 1F, in which endogenous NuMA is replaced with full-length NuMA either wild-type or 5A-3, the cortical clustering does not seem very evident.

Under physiological condition, cortical NuMA signals at metaphase are weak, and thus it is difficult to see punctate signals, especially when NuMA was visualized with the less intense mCherry. However, when we visualize endogenous NuMA with the brighter mClover, we repeatedly observed clear punctate NuMA signals as shown in Figure 7—figure supplement 1B-C. Because we think these residues mediate self-oligomerization of NuMA, for future work will conduct in vitro reconstitution of NuMA and selected mutants to visualize their structure as described by Harborth et al., 1999. However, this work is beyond the current paper.

- In the Discussion, the authors propose that NuMA clustering works within a DDN network to promote end-on attachment of astral microtubules to the cortex. To my knowledge, no compelling evidence for end-on attachment in this process has been provided. This model, although possible, tends to reflect what known for the attachment of microtubules to kinetochore. Therefore, in the absence of specific evidence, it might be preferable to down-state the model.

By analyzing the dynamics of EB3 and tubulin, Kozlowski et al., demonstrated that astral microtubule ends tend to interact with the cell cortex through an end-on configuration in *C. elegans* embryos. Similar results were reported in metaphase human cells by Samora et al. and Kwon et al. Thus, we propose that astral microtubule tips are captured by cortical force-generating machinery as an end-on configuration. However, as suggested by the reviewer, it is still unclear how the DDN clusters interact with astral microtubule plus-ends compared to the more advanced understanding of plus end interactions at kinetochores. Thus, according to the suggestion, we have reworded our model to provide caution as indicated below, and have highlighted that the ring-like structure for a NuMA cluster is an imaginary structure based on Harborth et al., in Figure 7—figure supplement 1G.

“… and astral microtubules tend to interact with the cell cortex through an end-on configuration in pre-anaphase cells (Kozlowski, Srayko, and Nedelec, 2007; Kwon, Bagonis, Danuser, and Pellman, 2015; Samora et al., 2011), it is tempting to speculate that the DDN cluster encircles or partially wraps the plus tip of a single astral microtubule …”

“The ring-like structure for a NuMA cluster is an imaginary structure based on Harborth et al., (Harborth et al., 1999).”

Reviewer #3:1) The idea that NuMA may activate dynein in addition to regulating its localization is an attractive hypothesis, which has been proposed by others but not carefully tested. Similarly, this work includes no direct data related to dynein activation. The authors show that the Spindly-like motif is required for dynein recruitment, but they do not decouple recruitment (localization) and activation. Thus, the use of language throughout the paper referring to dynein activation could be misleading; it should be softened and mostly confined to the Discussion. It should be made clear that the authors did not test whether NuMA activates dynein.

We appreciate this comment. To answer this question, we have now tested the effect of ciliobrevin D, an inhibitor of dynein activity, on spindle displacement caused by light-induced NuMA targeting. We carefully optimized the experimental conditions and found that ciliobrevin D treatment disrupts spindle displacement following cortical NuMA targeting. This result suggests that NuMA activates dynein. This data is now included in Figure 3A-D and Figure 3—figure supplement 1A-G.

2) In the experiments in Figures 4 and 5, why is NuMA 1-2115 ΔNLS used instead of the full-length protein? In the subsection “A Spindly-like motif in NuMA is required for cortical dynein recruitment, but not sufficient for spindle pulling”, the authors state that the NLS is removed to prevent dimerization with endogenous protein, but do not state a source. Do they believe this to be true, and due to what published or unpublished data? Published work (e.g., Harborth et al., EMBO 1995) suggests that dimerization will still occur through NuMA's coiled-coils.

As also described above in response to reviewer #2:

We removed the NLS to spatially separate exogenously expressed NuMA constructs from endogenous NuMA, which dominantly localizes in the nucleus of interphase cells. In fact, based on our synchronized expression protocol as described in Figure 4—figure supplement 1B, exogenously expressed NuMA constructs localized in cytoplasm before G2 release (at 68 hr), and separated from endogenous NuMA in the nucleus, suggesting that large majority of exogenously expressed NuMA mutants form homo-dimers through coiled-coil in the cytoplasm, but not with endogenous NuMA.

“… the NLS was deleted to eliminate dimerization with endogenous NuMA by spatially separating the exogenously expressed constructs from nuclear-localized endogenous NuMA before G2 release.”

In addition, we have performed these experiments with NLS containing full-length protein. In this case, exogenously expressed NuMA-RFP-Nano accumulated in the nucleus before G2, but was unable to displace the spindle efficiently (11.1%, n=9) during mitosis. We think that exogenously expressed full length NuMA forms heterodimers with endogenous NuMA lacking RFP-Nano, which leads to an inability of force generation due to weaker cortical anchorage. We have now added following comments to the text:

“In contrast, exogenously expressed NLS containing NuMA-RFP-Nano (1-2115) accumulated in the nucleus before G2, but was unable to displace the spindle efficiently (11.1%, n=9), likely due to weak cortical anchorage by hetero-dimerization with endogenous NuMA lacking RFP-Nano.”

3) Relatedly, dimerization with endogenous NuMA presumably occurs with NuMA truncations in Figures 4 and 5 that include some or all of the central coiled-coil. Could the authors comment on how this dimerization could affect their conclusions based on these data? As one example, could the inability of fragment #12 reflect an inability to form a functional dimer with endogenous protein due to absence of big part of the central coiled-coil?

All constructs except for NuMA-C #13 localized to the cytoplasm prior to G2 release in our synchronized protocol. Thus, we believe that these exogenously expressed constructs are spatially separated from endogenous NuMA before entering mitosis, and thus the majority of exogenously-expressed NuMA would not form heterodimers with endogenous NuMA during mitosis. NuMA-C #13 localized to the nucleus before G2 release, but this construct lacks the coiled-coil region required for dimerization, and thus we think that this construct does not dimerize with endogenous NuMA.

4) In Figure 4C and 5B, the application of a z-test to "yes/no" data (not normally distributed) with low "N" is likely inappropriate. We suggest using the actual pole displacement distances (as a percentage of starting pole-to-cortex distance) for each condition, and then analyzing those data using an ANOVA (or, at least, a t-test between the control (#1) and each condition). Consulting a statistician may be helpful.

According to this suggestion, we have now plotted the actual spindle displacement distance as a percentage of starting pole-to-cortex distance for each condition in Figure 4—figure supplement 1J. Because these samples do not show a clear Gaussian distribution, we performed a Mann-Whitney test, which can be applicable for such samples with low N (N>4). Although #2 and #11 are not significantly different, other samples show statistical difference (p<0.05) compared to control (#1). This data is now included in Figure 4—figure supplement 1J and mentioned in the figure legend.

5) Relatedly, directly reporting the actual pole displacement distances for NuMA fragments #1-14 (Figure 4-5) would be interesting and important for the reader to interpret the data. These data could be displayed as "beeswarm" plots like Figure 7C, for example, and they would help the reader better interpret the significance of your findings (for example, it would better display the degree of spindle displacement difference between NuMA fragment #11 (3/8) and fragment #14 (1/13)).

According to this suggestion, the data is now included in Figure 4—figure supplement 1J.

6) In model Figure 7D (and related Figure 7—figure supplement 1), the main features of the cartoon should stay as close to what was shown in the paper as possible. If there is key speculation not supported by data, it should be made clear that it is speculation. In particular, whether LGN forms a ring does not seem to be based on data and drawing LGN and NuMA's C-terminus as more disorganized may avoid readers being biased towards a ring-like model (alternatively, if a ring-like model is supported by data, it should be made clear what data supports it).

We have now commented that the ring-like structure is imaginary based on Harborth et al., 1999. In addition, we have included a non-ring structure model in Figure 7—figure supplement 1G. Because we found that LGN also displays punctate signals like NuMA, we drew LGN together with NuMA. This data is now included in Figure 7—figure supplement 1C.

“The ring-like structure for a NuMA cluster is an imaginary structure based on Harborth et al., (Harborth et al., 1999).”